# ULTra: Unveiling Latent Token Interpretability
# in Transformer-Based Understanding and Segmentation

**Hesam Hosseini**                                                            *hesam8hosseini@gmail.com*
*Sharif University of Technology*

**Ghazal Hosseini Mighan**                                                        *ghazaldesu@gmail.com*
*Sharif University of Technology*

**Amirabbas Afzali**                                                              *amir8afzali@gmail.com*
*Sharif University of Technology*

**Sajjad Amini**[*]                                              *s_amini@sharif.edu, samini@umass.edu*
*Sharif University of Technology*

**Amir Houmansadr**                                                               *amir@cs.umas.edu*
*University of Massachusetts Amherst*

**Reviewed on OpenReview:** *https://openreview.net/forum?id=vL3pmJjGDQ*

## Abstract

Transformers have revolutionized Computer Vision (CV) through self-attention mechanisms. However, their complexity makes latent token representations difficult to interpret. We introduce `ULTra`, a framework for interpreting Transformer embeddings and uncovering meaningful semantic patterns within them. `ULTra` enables unsupervised semantic segmentation using pre-trained models without requiring fine-tuning. Additionally, we propose a self-supervised training approach that refines segmentation performance by learning an external transformation matrix without modifying the underlying model. Our method achieves state-of-the-art performance in unsupervised semantic segmentation, outperforming existing segmentation methods. Furthermore, we validate `ULTra` for model interpretation in both synthetic and real-world scenarios, including Object Selection and interpretable text summarization using LLMs, demonstrating its broad applicability in explaining the semantic structure of latent token representations. [1]

## 1 Introduction

In recent years, the Transformer architecture and foundation models, which leverage self-attention mechanisms to capture complex dependencies, have transformed Natural Language Processing (NLP) benchmarks (Vaswani et al., 2017; Touvron et al., 2023; Team et al., 2024). Similarly, Vision Transformers (ViTs) (Dosovitskiy et al., 2020) have been adapted in Computer Vision (CV) and now serve as the backbone for various tasks such as segmentation and object detection (Thisanke et al., 2023; Liu et al., 2021). Despite their success, understanding the interpretability of Transformers remains a challenge due to the complexity of their latent token representations.

Several methods have been developed to enhance the interpretability of CNN-based models (Simonyan et al., 2014; Zeiler & Fergus, 2014; Selvaraju et al., 2017). While some of these can be extended to Transformer architectures, they do not fully leverage the unique attention mechanisms inherent to Transformers. Recent

---

[*]This work was conducted during a visit at CICS, UMass Amherst.
[1]The codebase is available at https://github.com/CocoAika/ULTra

research has introduced interpretability methods specifically designed for Transformers (Chefer et al., 2021b; Abnar & Zuidema, 2020; Vig & Belinkov, 2019). However, these approaches primarily aim to explain the final outputs of a model and cannot be directly applied to interpret latent tokens, offering only limited insight into the intermediate processes that drive predictions. This gap naturally raises a fundamental question:

### *Do transformer models exhibit semantic awareness within their latent representations?*

To address this, we propose *Unveiling Latent Token Interpretability in Transformer-Based Understanding (ULTra)*, a framework for interpreting latent tokens in Transformers. At a high level, `ULTra` interprets latent tokens by tracing how input information flows through the Transformer's attention layers to individual token representations. Given a pre-trained Transformer and a target layer, we select a latent token and define a scalar function of its embedding. We then backpropagate this signal through the attention probability matrices, constructing layer-wise contribution maps that are aggregated across layers to produce a token-specific explanation map in the input space. These explanation maps reveal which input regions most strongly influence each latent token, exposing their semantic specialization without modifying the model or requiring aligned modalities.

Recent work with similar objectives (Chen et al., 2024) interprets latent tokens by projecting them into CLIP's multi-modal embedding space. This is achieved by disabling the self-attention mechanism and then using an external text encoder to associate tokens with semantic descriptions. In contrast, `ULTra` (i) offers a direct interpretation of latent tokens without relying on any aligned modality, and (ii) preserves the model architecture and behavior at inference time, thereby avoiding modifications that could compromise faithfulness.

By examining the semantic information encoded in latent tokens, we demonstrate that Transformers inherently capture the semantic structure of their input as a collection of distinct concepts. For instance, in Figure 2, tokens clearly separate semantic entities, with individual tokens specializing in the dog, the cat, the background, or even fine-grained attributes such as the cat's head. This observation naturally leads to a second question:

### *Can such alignment be exploited for unsupervised semantic segmentation (USS)?*

we cluster token-level explanation maps and aggregate them to form pixel-wise semantic regions, thereby achieving unsupervised semantic segmentation. Unlike prior unsupervised segmentation methods that require additional training (Sick et al., 2024; Hamilton et al., 2022; Li et al., 2023), `ULTra` leverages the intrinsic knowledge of pre-trained models on tasks other than semantic segmentation to achieve state-of-the-art performance on benchmark datasets without the need for fine-tuning. To further enhance segmentation performance, we introduce a self-consistency approach that learns an external transformation matrix in a self-supervised manner, refining segmentation without modifying the underlying model. Additionally, we validate our interpretability framework on transformer-based LLMs through qualitative analyses in text summarization, demonstrating its broad applicability across modalities.

Our main contributions are as follows:

- We propose a framework for interpreting latent tokens in Transformers, uncovering the stored semantic knowledge in each latent token. To the best of our knowledge, we are the first to investigate latent token interpretability directly.

- We extend `ULTra` to several designed synthetic and real-world tasks, showcasing its versatility and applicability, including Object Selection and interpretation of LLMs in text summarization tasks.

- By aggregating explanation maps generated from latent tokens, our method enables unsupervised semantic segmentation using pre-trained ViTs without requiring any additional training. This strategy outperforms existing state-of-the-art approaches that rely on fine-tuning or supervision, highlighting the effectiveness of leveraging the semantic structure embedded in large-scale models.

- To further enhance segmentation performance, we introduce $\text{ULTra}_{\mathcal{W}}$, a lightweight learnable extension that optimizes a self-consistency loss. This loss encourages stable token representations under

input perturbations and yields a transformation matrix that projects tokens onto a more informative subspace while keeping the backbone model unchanged.

## 2 Related Work

**Interpretable Deep Learning.** Model interpretability is a critical aspect of deep learning, particularly for complex architectures like Transformers. Traditional methods such as saliency maps (Simonyan et al., 2014), Grad-CAM (Selvaraju et al., 2017), LIME (Ribeiro et al., 2016), and SHAP (Lundberg, 2017) have been effective for CNNs but do not fully exploit Transformers' self-attention mechanisms. A growing body of research focuses on Transformer-specific interpretability techniques that leverage attention mechanisms as intrinsic explanations (Vig & Belinkov, 2019; Abnar & Zuidema, 2020; Chefer et al., 2021b; Jain & Wallace, 2019; Wu et al., 2024). These methods are largely post-hoc, providing explanations after model training. In contrast, ante-hoc approaches such as IA-ViT (Qiang et al., 2023) and Mechanistic Interpretability (Rai et al., 2024) seek to make Transformers inherently interpretable. Additionally, techniques like LeGrad (Bousselham et al., 2024) and IA-RED$^2$ (Pan et al., 2021) improve interpretability by analyzing feature formation and reducing redundancy in self-attention. Despite these advancements in model interpretability, understanding latent tokens in ViTs remains underexplored, underscoring the need for methods that explicitly interpret these latent tokens. A related study (Chen et al., 2024) explores latent token interpretation in CLIP by modifying self-attention mechanisms.

**Semantic Segmentation.** Unsupervised semantic segmentation has progressed through self-supervised learning and clustering techniques. Early methods such as IIC Ji et al. (2019) and PiCIE Cho et al. (2021) leveraged mutual information and consistency principles to enhance feature representations. Transformer-based approaches like DINO Caron et al. (2021) and STEGO Hamilton et al. (2022) further improved segmentation by capturing meaningful structures through self-attention. Other techniques, including MaskContrast Van Gansbeke et al. (2021), Leopart Ziegler & Asano (2022), and ACSeg Li et al. (2023), refined segmentation through clustering and adaptive conceptualization. More recent methods, such as DepthG Sick et al. (2024), incorporate depth-guided correlations, while U2SEG Niu et al. (2024) utilizes pseudo-labeling. Additionally, SmooSeg Lan et al. (2023) enforces smoothness priors, and HSG Ke et al. (2022) applies hierarchical segmentation via multiview clustering Transformers. In the context of semantic segmentation, Weakly Supervised Semantic Segmentation (WSSS) aims to generate segmentation masks using only image-level labels. These methods often rely on saliency maps from interpretability techniques, such as Class Activation Maps (CAMs), to localize objects Choe & Shim (2019); Yin et al. (2020); Chen et al. (2023). Typically, they depend on class logits to guide segmentation, refining masks based on predicted class scores.

**Latent Embedding.** The high dimensionality and complex distribution of latent embeddings in deep models pose significant challenges for interpretation and manipulation. Methods like GroupViT Xu et al. (2022) utilize hierarchical grouping to facilitate more meaningful representation learning, enabling semantic segmentation. Other approaches, such as Lee et al. (2024); Bolya et al. (2023); Liang et al. (2022), improve computational efficiency by eliminating redundant tokens, while register-based ViTs Darcet et al. (2024) address artifacts in feature maps caused by outlier-norm tokens through the introduction of register tokens. However, these techniques primarily emphasize computational efficiency and representation structuring rather than the interpretability of latent embeddings.

## 3 Methodology

In this section, we introduce our approach for interpreting latent representations in Transformers. We start by outlining the essential preliminaries, followed by a detailed explanation of the `ULTra` framework for analyzing latent tokens. Additional details on the Transformer architecture are provided in Appendix I.

### 3.1 Preliminaries

Previous research on attention-based model interpretability (Abnar & Zuidema, 2020; Chefer et al., 2021b;a; Wu et al., 2024) has primarily focused on analyzing the semantic flow from input tokens to class logits.

Figure 1: The overall architecture of the `ULTra` framework. The framework consists of a forward path, where the input data $\mathbf{x}$ is fed into the model, and a backward path, starting from the target layer $l$, where we compute the gradient of a scalar function of the $i$-th latent token, $f(\mathbf{z}_i^{(l)})$, with respect to the attention probability matrix of the middle layer $b$. Next, we compute the corresponding contribution map $\mathbf{C}_i^{(b,l)}$ for all middle layers. Finally, we construct the explanation map $\overline{S}_i^{(l)}$, select its $i$-th row, and transform it to the input size. As an example, on the left, we observe that the token corresponding to the middle window assigns considerable attention to the left window, suggesting an underlying semantic understanding.

This is typically achieved by adding the attention probability matrix of each layer to an identity matrix and aggregating the results across attention heads through a weighted average. The weights are derived from the gradients of the class logits with respect to the attention probabilities. Specifically, to assess the contribution of each class logit using attention information, the contribution map for layer $b$ is defined as:

$$\mathbf{C}_c^{(b)} = \mathbf{I} + \mathbb{E}_h \left[ \left( \nabla_{\mathbf{A}_h^{(b)}} p(c) \right)^+ \odot \mathbf{A}_h^{(b)} \right], \tag{1}$$

$$\overline{S}_c = \mathbf{C}_c^{(1)} \cdot \mathbf{C}_c^{(2)} \cdots \mathbf{C}_c^{(L)}, \tag{2}$$

$$S_c = \overline{S}_c[0, 1 :], \tag{3}$$

where $\odot$ denotes the Hadamard product, and $(\cdot)^+$ is the operator that retains only positive values. $\mathbf{I}$ represent the identity matrix, which reflects the contributions of skip connections and $\mathbf{A}_h^{(b)} \in \mathbb{R}^{(n+1) \times (n+1)}$ denotes the attention probability matrix of head $h$ and layer $b$. The term $\nabla_{\mathbf{A}_h^{(b)}} p(c) = \frac{\partial p(c)}{\partial \mathbf{A}_h^{(b)}}$ is the partial derivative of the attention map with respect to the predicted probability for class $c$, and $\mathbb{E}_h$ denotes the mean over multiple attention heads. Equation 2 represents matrix multiplication, which accounts for aggregating contributions from all layers. Each element of $\overline{S}_c[i, j]$ represents the influence of the $j$-th input token on the $i$-th output token. Thus, each element of $S_c[j]$ quantifies the influence of the $j$-th input token on class $c$, with token 0 corresponding to the `CLS` token. This raises a new question: ***Can we expect similar semantically meaning for a latent token? Is there a similar approach to interpret latent tokens in Transformers?***

### 3.2 ULTra Framework

Inspired by Equation 1, in this work we aim to measure the contribution of latent token $\mathbf{z}_i^{(l)}$ to the input space using the underlying attention probabilities, where $i \in \{0, 1, \ldots, n\}$ denotes the token index and $l$ represents the layer. Specifically, to quantify the influence of the attention map at layer $b$ on the latent token $\mathbf{z}_i^{(l)}$, the contribution map $\mathbf{C}_i^{(b,l)}$ is defined as follows:

$$\mathbf{C}_i^{(b,l)} = \mathbf{I} + \mathbb{E}_h \left( \left( \nabla_{\mathbf{A}_h^b} f(\mathbf{z}_i^{(l)}) \right)^+ \odot \mathbf{A}_h^b \right), \tag{4}$$

Intuitively, this approach traces the most influential input information contributing to a latent token $\mathbf{z}_i^{(l)}$, where the notion of *influence* is defined based on the impact an input token has on the function $f : \mathbb{R}^n \to \mathbb{R}$ of the target token. Specifically, by *influence*, we refer to the degree to which an input token affects $f(\mathbf{z}_i^{(l)})$, reflecting its relative importance in the representation. Since tokens are high-dimensional vectors, assuming equal importance across all components can obscure critical contributions. Moreover, different aggregation strategies can assign varying importance to individual dimensions. To address this, we evaluated three functions: (i) a simple summation over all components, (ii) the vector norm (treating all components equally), and (iii) our proposed ULTra$_{\mathcal{W}}$, which learns a weighting function. Empirically, ULTra$_{\mathcal{W}}$ generally yields stronger downstream performance.

Finally, we define the corresponding explanation map for the latent token $\mathbf{z}_i^{(l)}$, denoted as $S_i^{(l)} \in \mathbb{R}^n$, where each element represents the influence of an input token on $\mathbf{z}_i^{(l)}$. The explanation map is computed as:

$$\overline{S}_i^{(l)} = \mathbf{C}_i^{(1,l)} \cdot \mathbf{C}_i^{(2,l)} \cdots \mathbf{C}_i^{(l-1,l)}, \quad S_i^{(l)} = \overline{S}_i^{(l)}[i, 1 :], \tag{5}$$

where . represents matrix multiplication, which aggregates contributions from all layers to the target latent token. Transformer skip connections cause most contributions to concentrate on $S_i^{(l)}[i-1]$, hindering token-level analysis. To mitigate this, we replace it with the maximum of the other elements, better capturing token contributions. Moreover, in some experiments, e.g., semantic segmentation, we reshape and upsample the explanation map using bilinear or cubic interpolation to match the input resolution, producing $\tilde{S}_i^{(l)}$. The overall framework of `ULTra` is shown in Figure 1.

Additionally, As we illustrate in section 5, in our experiments we utilize three approaches for designing $f$ in `ULTra` framework:

**(i)** $f_{\mathbf{s}}(\mathbf{z}_i^{(l)}) = \langle \mathbf{z}_i^{(l)}, \mathbf{1} \rangle$ In this formulation, $\mathbf{1}$ represents an all-ones vector, meaning that the function simply computes the sum of all elements in $\mathbf{z}_i^{(l)}$. This approach is denoted by ULTra$_{\mathcal{S}}$. The underlying intuition is that this approach treats all elements of the token equally.

**(ii)** $f_{\mathbf{e}}(\mathbf{z}_i^{(l)}) = \langle \mathbf{z}_i^{(l)}, \mathbf{z}_i^{(l)} \rangle$ This approach is based on the energy (or norm) of a given token, a concept that has been previously explored in the literature Darcet et al. (2024). By leveraging token energy, this method captures the magnitude of the token's latent representation. We refer to this variant as ULTra$_{\mathcal{E}}$.

**(iii)** $f_{\mathbf{w}}(\mathbf{z}_i^{(l)}) = \langle \mathbf{z}_i^{(l)}, \mathbf{w}_i \rangle$: Beyond the previous approaches, we introduce a learnable vector $\mathbf{w}_i$ to project each token representation $\mathbf{z}_i^{(l)}$. Geometrically, the function $f_{\mathbf{s}}$ in ULTra$_{\mathcal{S}}$ can be viewed as mapping all tokens onto a shared projection basis within the token space. In contrast, learning $\mathbf{w}_i$ enables the construction of a more informative, robust projection basis. We refer to this variant as ULTra$_{\mathcal{W}}$. Unlike ULTra$_{\mathcal{S}}$ and ULTra$_{\mathcal{E}}$, which are entirely training-free, ULTra$_{\mathcal{W}}$ requires light training to optimize the $\mathbf{w}_i$ vectors. In practice, we train these vectors using only a limited number of batches, yet this additional flexibility improves segmentation performance. In Section 4.1, we discuss this idea more in detail by introducing a self-supervised strategy for learning $\mathbf{w}_i$, which relies on another `ULTra` variants for an initial segmentation.

For both ULTra$_{\mathcal{S}}$ and ULTra$_{\mathcal{E}}$, our framework remains fully training-free. In semantic segmentation tasks, experimental results show that these variants not only approach but often surpass baselines that rely on fine-tuning, underscoring both their effectiveness and efficiency.

| (a) Original Image | (b) Latent Token's explanation Map | (c) Predicted Binary Mask |
| --- | --- | --- |

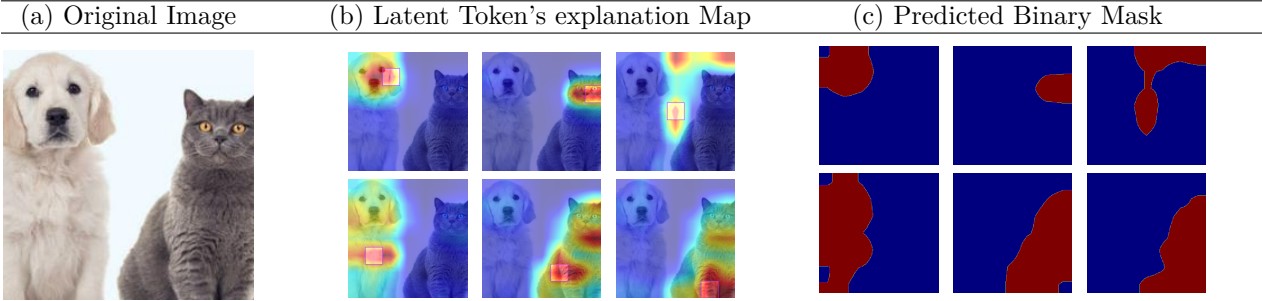

Figure 2: An example of token interpretation by our model and its predicted binary mask. (a) Original image. (b) Overlay of $\tilde{S}_i^{(13)}$ on the original image for different $i$, where the location of the $i$-th token is indicated by the purple square. (c) The binary mask $M_i^{(13)}$ for each corresponding explanation map in (b). We can generally observe that tokens clearly separate semantic entities, attending to the dog, the cat, the background, or even fine-grained attributes like the cat's or dog's head.

## 4 Tailoring ULTra for Different Applications

In this section, we aim to examine ULTra's capability to adapt to various tasks involving semantic knowledge.

### 4.1 Unsupervised Semantic Segmentation

Explanation maps are defined for each latent token at a fixed layer, resulting in as many maps as there are latent tokens. For segmentation, we use hierarchical clustering to group these maps. This clustering method naturally matches how vision transformers process images, eventually aggregating similar concepts at multiple scales of fineness. It finds the structure in token explanations without needing us to guess the number of clusters beforehand. In Appendix G Figure 14 also provides a schematic of this process. In our experiments, we set a predefined number of clusters $k$ for the algorithm which setup the threshold corresponding to that number of classes, however as we discuses in the appendix G segmenting using only the threshold $\zeta$ gives flexible control over segmentation detail; lower values show fine details and higher values create broader regions. To ensure fair representation, we apply Min-Max scaling to prevent larger objects from dominating the clustering process.

After clustering, $k$ distinct concepts are defined by aggregating explanation maps. The aggregated explanation map for cluster $c$ is:

$$\tilde{S}_c^{(l)}[x,y] = \sum_{i \in \phi(c)} \tilde{S}_i^{(l)}[x,y], \qquad (6)$$

where $\phi(c) = \{i : \text{Class}(\tilde{S}_i^{(l)}) = c\}$ represents the grouping of label assignments. Class labels are assigned to input pixels using the explanation map of the $l$-th layer as follows:

$$\text{Class}[x,y]^{(l)} = \underset{c \in \{1,\ldots,k\}}{\operatorname{argmax}} S_c^{(l)}[x,y]. \qquad (7)$$

Some examples illustrating our segmentation method are presented in Figure 3.

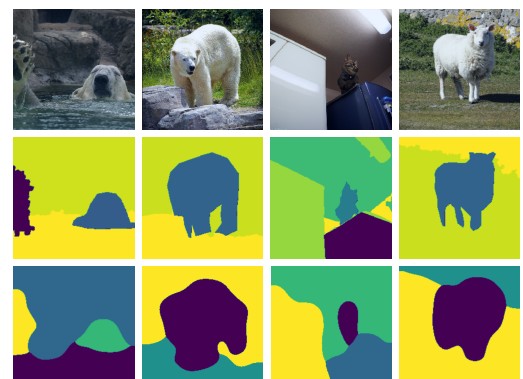

Figure 3: ULTra segmentation results on sample images. The top row displays the original images, the middle row shows true annotations, and the bottom row presents our model's predictions.

**ULTra$_\mathcal{W}$.** As previously mentioned, to further enhance segmentation performance, we propose ULTra$_\mathcal{W}$, which utilizes a learnable transformation matrix to obtain a more informative projection basis for the token space. For simplicity of notation, we fix the layer $\ell$ and omit the layer superscript in the following derivations. Consider $z_{i,j} = \mathbf{e}_j^T \mathbf{z}_i$, where $\mathbf{e}_j$ is the $j$-th standard basis vector. The gradients can then be rewritten as:

$$\nabla_{\mathbf{A}_h^b} f_{\mathbf{s}}(\mathbf{z}_i) = \sum_j \nabla_{\mathbf{A}_h^b} z_{i,j}, \quad \nabla_{\mathbf{A}_h^b} f_{\mathbf{w}}(\mathbf{z}_i) = \sum_j \mathbf{w}_{i,j} \nabla_{\mathbf{A}_h^b} z_{i,j}. \tag{8}$$

In other words, the gradient of $f_{\mathbf{w}}(\mathbf{z}_i)$ is a weighted version of the gradient of $f_{\mathbf{s}}(\mathbf{z}_i)$. From this perspective, by manipulating the weighting vector $\mathbf{w}_i$, we can prioritize the more informative dimensions of the token space. As illustrated in Section 5, this leads to improved performance of ULTra compared to variants that rely on a naive choice of $f$.

Here, a crucial question arises: *How do we characterize the amount of information in the token space to achieve a good projection basis for the token space?* In other words, we need to learn a matrix $\mathbf{W} = [\mathbf{w}_1, \ldots, \mathbf{w}_n]^T$ to perform more effective segmentation. To this end, we introduce a perturbation on the input image and, by optimizing $\mathbf{W}$, aim to minimize the divergence of specific tokens from their unperturbed versions. This motivation is supported by prior findings showing that transformers remain robust under noisy inputs Zhou et al. (2022), suggesting that they encode information in noise-resistant representations.

Let $\mathbf{x}$ be an input image and $\tilde{\mathbf{x}}$ its perturbed version corresponding to segmentation class $c$, which can be computed as:

$$\tilde{\mathbf{x}}^c = \mathbf{x} + \mathcal{P}_\phi(\mathbf{x}, c; \delta), \tag{9}$$

where $\mathcal{P}_\phi{}^2$ is a perturbation strategy that applies perturbations to the region that are not predicted for class $c \in \mathcal{C}$ using the initial segmentation. Moreover, $\delta$ quantifies the amount of perturbation, and for $\delta = 0$, we have $\tilde{\mathbf{x}}^c = \mathbf{x}$.

Now, to optimize the parameters $\mathbf{W}$, we formulate the problem as:

$$\min_{\mathbf{W}} \mathbb{E}_{x \sim \mathcal{D}} \left[ \sum_{c \in \mathcal{C}} \sum_{i \in \{j | \mathbf{x}_j = \mathbf{x}_j^c\}} d_{\mathbf{W}}(\mathbf{z}_i, \tilde{\mathbf{z}}_i) \right], \quad \text{s.t.} \quad \|\mathbf{w}_k\|_2 = 1, \ \forall k \in \{1, \cdots, n\}, \tag{10}$$

where $d_{\mathbf{W}}(\mathbf{z}_i, \tilde{\mathbf{z}}_i)$ denotes the divergence between the two vectors based on the projection basis $\mathbf{w}_i$, $\|\cdot\|_2$ is the Euclidean norm, and $\mathbf{x}_i$ and $\mathbf{z}_i$ represent the $i$-th input token and latent token, respectively. In particular, we do not perturb the regions recognized by ULTra$_{\mathcal{W}}$ that contain localized information. This encourages the model to become more aware of self-attended regions, producing sharper and more focused heatmaps over positively contributing areas. Consequently, the model's decision-making becomes more precise in these critical regions. Qualitative examples and additional discussion of ULTra$_{\mathcal{W}}$ are provided in Appendix C.

In our experiments, we parameterize each projection vector as $\mathbf{w}_k = \boldsymbol{\theta}_k / \|\boldsymbol{\theta}_k\|_2$. Hence, based on the formulation in Equation 10, to optimize the parameters $\boldsymbol{\theta}_k$, we define the *Self-Consistency* loss as follows:

$$\mathcal{L}_{\text{sc}}(x, \boldsymbol{\Theta}) = \sum_{c \in \mathcal{C}} \sum_{i \in \{j | \mathbf{x}_j = \mathbf{x}_j^c\}} \left| \mathbf{w}_i^T (\mathbf{z}_i - \tilde{\mathbf{z}}_i) \right|^2, \tag{11}$$

where $\mathbf{w}_i = \boldsymbol{\theta}_i / \|\boldsymbol{\theta}_i\|_2$ and $\boldsymbol{\Theta} = [\boldsymbol{\theta}_1, \ldots, \boldsymbol{\theta}_n]^T$. We use the term *"Self"* because we leverage the model's own segmentation results, obtained using ULTra$_{\mathcal{W}}$, to enhance its performance through a systematic weighting approach. Experimentally, we found that lightly optimizing $\boldsymbol{\Theta}$ on a limited number of samples improves the model's segmentation performance.

## 4.2 Latent Token Interpretability Assessment

A key challenge in interpretability is the lack of a universal evaluation metric Chefer et al. (2021b); Wu et al. (2024). In supervised settings, explainability is typically evaluated indirectly via downstream tasks (e.g., semantic segmentation) or perturbation tests. Assuming strong explanation maps, performance should improve or coherent output changes should occur under perturbation. Since tokens in ULTra naturally have no labels, we extend this principle to the unsupervised domain. As we already discussed how unsupervised semantic segmentation serves as an interpretability evaluation approach for `ULTra`, in this section we evaluate `ULTra` using additional introduced metrics. Specifically, we designed two tasks: (i) Perturbation Test and

---

[2]The perturbation strategy is defined based on the model parameters $\phi$ since the model is involved in distinguishing the perturbation using its segmentation.

| (a) Original Image | (b) Explanation Relevency Map |
|---|---|

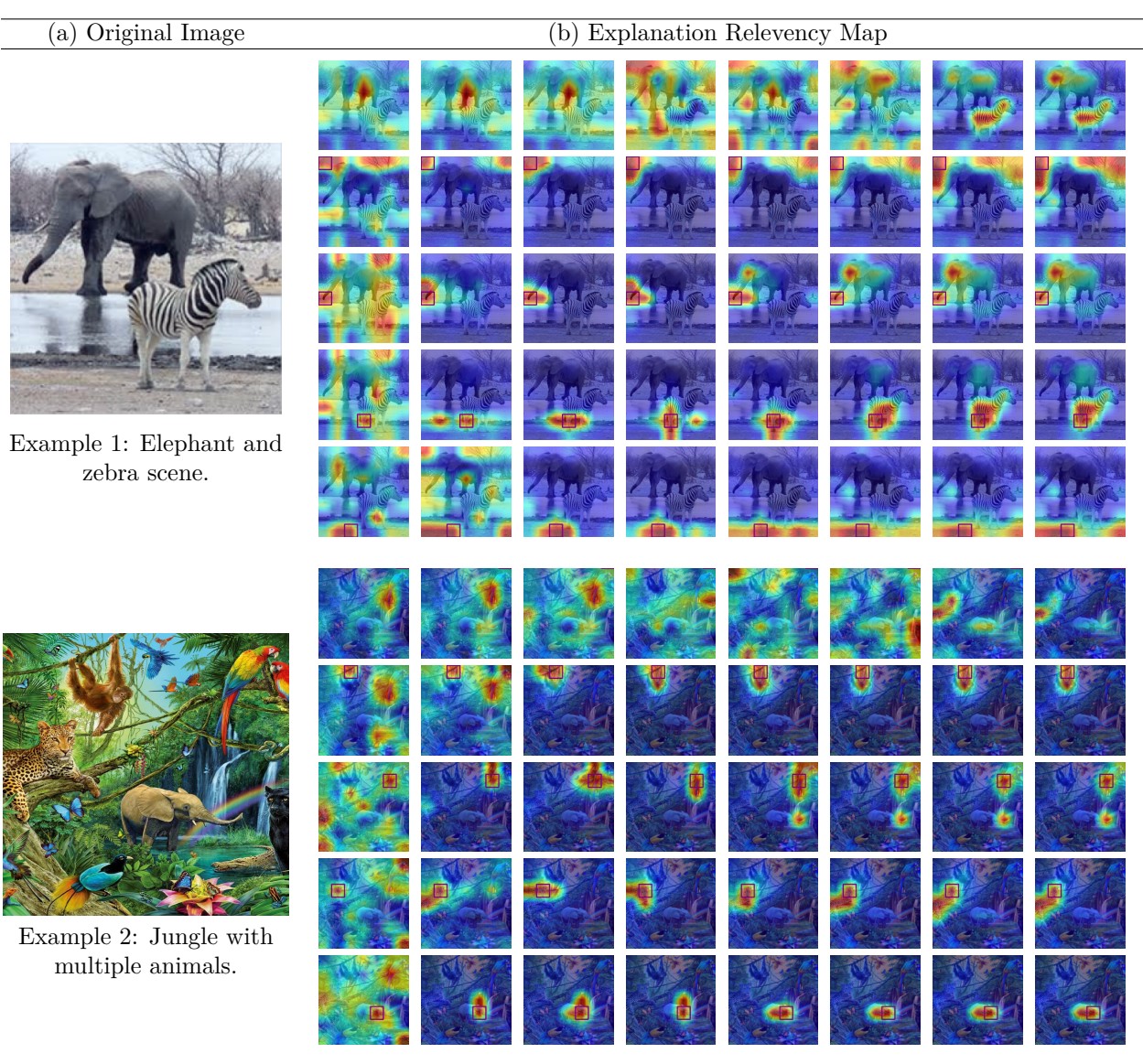

Figure 4: Two examples illustrating the model's decision-making across layers. Columns correspond to progressively deeper layers. The first row for each image shows the `CLS` token, while the subsequent rows show three selected tokens (highlighted by red squares). Deeper layers capture richer semantics: in Example 1, elephant, zebra, and background tokens become increasingly distinct as we go through layers. The `CLS` token represents both animals but does not differentiate background types (sky, ground, or water), whereas token-level representations do. In Example 2, which contains multiple animals, the `CLS` token captures only the tiger and part of the elephant, while other tokens represent additional objects. Interestingly, the model confuses the parrot with part of the rainbow due to similar colors.

(ii) Object Selection. Additionally, in Section 5.3, we propose quantitative metrics to evaluate our method on these tasks to provide deeper insights into the interpretability of `ULTra` by identifying influential regions and behaviors within the model's decision-making process.

**Perturbation Test**. The perturbation test is a widely used metric for evaluating explainability methods Chefer et al. (2021b); Wu et al. (2024). This approach involves perturbing parts of an image based on the explanation map generated by the method and measuring the resulting change in the predicted class probability. Inspired by this technique, instead of using class probability, we leverage the change in the

embedding space to design a perturbation-based validation test. This allows us to assess the reliability of the explanation maps generated for the latent tokens.

**Object Selection.** In this task, we convert the upsampled explanation map $\tilde{S}_i^{(l)}$ into a binary segmentation mask using a threshold $\tau$, where the binary mask $M_i^{(l)}$ is defined as:

$$M_i^{(l)}[x,y] = \begin{cases} 0, & \text{if } \tilde{S}_i^{(l)}[x,y] < \tau, \\ 1, & \text{otherwise.} \end{cases} \quad (12)$$

Here, $\tilde{S}_i^{(l)}[x,y]$ represents the relevance value at position $[x,y]$ in $\tilde{S}_i^{(l)}$, with $\tau$ as the threshold. When $M_i^{(l)}[x,y] = 1$, it indicates that the position $[x,y]$ belongs to the object region.

Our findings indicate that as tokens propagate through the network, they refine their object representation while retaining the semantic meaning of their associated image patches, performing *Object Selection*. Figure 4 visually illustrates this process. Deeper layers show richer semantic awareness, with tokens gradually capturing entire objects like the elephant or zebra, or the background, while the CLS token captures both the zebra and elephant. For a given patch token $\mathbf{x}_i$, the object it most strongly represents is denoted as class $k_i$. The latent token $\mathbf{z}_i^{(l)}$ generates an explanation map that highlights areas with higher values associated with class $k_i$ in the whole image, including $\mathbf{x}_i$. After applying a threshold, this map becomes a binary segmentation mask expected to exhibit a high Intersection over Union (IoU) with the corresponding class $k_i$ region in the image. An illustrative example is shown in Figure 2.

| Method | Model | Training | U. ACC | U. mIoU |
|---|---|:---:|---|---|
| IIC | R18+FPN | ✓ | 21.8 | 6.7 |
| PiCIE | R18+FPN | ✓ | 48.1 | 13.8 |
| DINO | ViT-S/8 | ✓ | 28.7 | 11.3 |
| | ViT-S/16 | ✓ | 22.0 | 8.0 |
| | ViT-B/8 | ✓ | 30.5 | 9.6 |
| ACSeg | ViT-S/16 | ✓ | - | 16.4 |
| TransFGU | ViT-S/8 | ✓ | 52.7 | 17.5 |
| STEGO | ViT-S/8 | ✓ | 48.3 | 24.5 |
| | ViT-S/16 | ✓ | 52.5 | 23.7 |
| | ViT-B/8 | ✓ | 56.9 | 28.2 |
| STEGO +HP | ViT-S/8 | ✓ | 57.2 | 24.6 |
| | ViT-S/16 | ✓ | 54.5 | 24.3 |
| DepthG | ViT-S/8 | ✓ | 56.3 | 25.6 |
| | ViT-B/8 | ✓ | 58.6 | 29.0 |
| U2Seg | R50 | ✓ | 63.9 | 30.2 |
| ULTra$_\mathcal{W}^{\text{CLIP}}$ | ViT-B/32 | ✓ | 60.6 | 34.1 |
| | ViT-B/16 | ✓ | 63.8 | 34.0 |
| | ViT-L/14 | ✓ | **67.9** | **38.2** |
| ULTra$_\mathcal{S}^{\text{CLIP}}$ | ViT-B/32 | ✗ | 60.8 | 34.6 |
| | ViT-B/16 | ✗ | 63.0 | 33.2 |
| | ViT-L/14 | ✗ | 66.5 | 37.5 |
| ULTra$_\mathcal{E}^{\text{CLIP}}$ | ViT-B/32 | ✗ | 59.5 | 32.6 |
| | ViT-B/16 | ✗ | 53.6 | 26.6 |
| | ViT-L/14 | ✗ | 59.0 | 31.7 |
| ULTra$_\mathcal{W}^{\text{DINO}}$ | ViT-S/16 | ✓ | 67.2 | 34.4 |
| | ViT-B/16 | ✓ | 67.4 | 37.7 |
| ULTra$_\mathcal{S}^{\text{DINO}}$ | ViT-S/16 | ✗ | 66.4 | 33.3 |
| | ViT-B/16 | ✗ | 67.3 | 35.6 |
| ULTra$_\mathcal{E}^{\text{DINO}}$ | ViT-S/16 | ✗ | 63.4 | 31.6 |
| | ViT-B/16 | ✗ | 63.0 | 31.3 |

Table 1: Comparison of unsupervised segmentation methods on the COCO-Stuff dataset.

### 4.3 Interpreting LLMs in Text Summarization

In this section, we examine how our interpretability framework can be applied to text summarization tasks, taking steps toward uncovering the underlying intent of LLMs. In Section 5.4, we qualitatively evaluate ULTra$_\mathcal{S}$ using two samples by visualizing the regions of the input context that an LLM prioritizes while interpreting a given TL;DR summary. This analysis reveals key input regions shaping the model's decisions, aiding in understanding how concise and relevant summaries are generated.

For this task, we concatenate the context $x$ and the summary $y$ with a separator token. After feeding this input into the model, we compute the relevance scores of the TL;DR tokens with respect to the context tokens.

We then average these scores for each token in $x$ to obtain a scalar value, referred to as the Token Contribution Score, $\lambda_i^{(l)} \in \mathbb{R}^+$, which highlights the contribution of each context token in interpreting the summary $y$ within the given context. Accordingly, $\lambda_i^{(l)}$ is computed as:

$$\lambda_i^{(l)} = \frac{1}{|y|} \sum_{j=1}^{|y|} S_{j+|x|}^{(l)}[i], \quad \forall i \in \{1, \cdots, |x|\},\tag{13}$$

where $|\cdot|$ denotes the number of tokens in the text.

## 5 Experiments & Results

### 5.1 Experimental Setup

**Datasets.** In our experiments, we evaluate model performance on several unsupervised semantic segmentation benchmarks, focusing on vision-related tasks. We conducted experiments on four datasets: COCO-Stuff 27 Caesar et al. (2018), PASCAL VOC 2012 Everingham & Winn (2011), Potsdam-3 ISPRS (2018), and Cityscapes Cordts et al. (2016).

For our qualitative analysis of LLM interpretation in the task of text summarization, as described in Section 4.3, we utilized the TL;DR dataset (Stiennon et al., 2022). This dataset contains summary comparisons with human feedback collected by OpenAI.

**Models.** For all experiments in the vision tasks, we used different pretrained versions of CLIP's image encoder (Radford et al., 2021) as well as DINO ViT-S/16 and ViT-B/16 (Caron et al., 2021). For interpreting text summarization, as described in Section 4.3, we used the Llama-2-7B language model (Touvron et al., 2023). All experiments were run on 4 NVIDIA A100-80GB GPUs.

Further details of the datasets and models used are provided in Appendix E.

### 5.2 Semantic Segmentation

To evaluate the effectiveness of our approach, we use the Unsupervised mean Intersection over Union (U. mIoU) and Unsupervised Pixel Accuracy (U. ACC) metrics. The experimental results on the COCO-Stuff, PASCAL VOC, Potsdam, and Cityscapes datasets are reported in Tables 1, 3, 4, and 2, respectively. In these tables, the Training column indicates whether any additional training is required. Moreover we present the result of using the threshold $\zeta$ instead of predefined number of class $k$ in the appendix G along with its sensitivity analysis.

We benchmarked the segmentation performance of our approach against several state-of-the-art (SOTA) methods in unsupervised segmentation. The projection matrix $W$ W in ULTra$_{\mathcal{W}}$ was optimized using the ADAM optimizer with a learning rate of 0.01 and a batch size of 32, over a total of 256 training samples and 10 epochs. This compact training setup suffices, as $W$ implements a linear mapping with a parameter count equal to the embedding dimensionality.

Among the proposed variants of ULTra, ULTra$_{\mathcal{W}}$ achieves the highest performance, requiring only a small number of training samples. Notably, even when no training data is available, ULTra$_{\mathcal{S}}$ still

| Method | Model | Training | U. mIoU |
|---|---|---|---|
| IIC | R18+FPN | ✓ | 9.8 |
| MaskContrast | R50 | ✓ | 35.0 |
| Leopart | ViT-S/16 | ✓ | 41.7 |
| TransFGU | ViT-S/8 | ✓ | 37.2 |
| MaskDistill | ViT-S/16 + R50 | ✓ | 42.0 |
| ACSeg | ViT-S/16 | ✓ | 47.1 |
| ULTra$_{\mathcal{W}}^{\mathrm{CLIP}}$ | ViT-B/32 | ✓ | **51.2** |
| | ViT-B/16 | ✓ | 50.9 |
| | ViT-L/14 | ✓ | 49.1 |
| ULTra$_{\mathcal{S}}^{\mathrm{CLIP}}$ | ViT-B/32 | ✗ | 49.2 |
| | ViT-B/16 | ✗ | 48.3 |
| | ViT-L/14 | ✗ | 48.7 |
| ULTra$_{\mathcal{E}}^{\mathrm{CLIP}}$ | ViT-B/32 | ✗ | 50.0 |
| | ViT-B/16 | ✗ | 40.0 |
| | ViT-L/14 | ✗ | 45.2 |
| ULTra$_{\mathcal{W}}^{\mathrm{DINO}}$ | ViT-S/16 | ✓ | 48.9 |
| | ViT-B/16 | ✓ | 50.5 |
| ULTra$_{\mathcal{S}}^{\mathrm{DINO}}$ | ViT-S/16 | ✗ | 47.9 |
| | ViT-B/16 | ✗ | 50.0 |
| ULTra$_{\mathcal{E}}^{\mathrm{DINO}}$ | ViT-S/16 | ✗ | 46.9 |
| | ViT-B/16 | ✗ | 50.0 |

Table 2: Comparison on the PASCAL VOC 2012 dataset. The Training column Indicates if fine-tuning is required

| Method | Model | Training | U. mIoU |
|---|---|---|---|
| IIC | R18+FPN | ✓ | 6.4 |
| PiCIE | R18+FPN | ✓ | 12.3 |
| STEGO | ViT-B/8 | ✓ | 21.0 |
| STEGO +HP | ViT-S/8 | ✓ | 18.4 |
|  | ViT-B/8 | ✓ | 18.4 |
| DepthG | ViT-B/8 | ✓ | 23.1 |
| ULTra$_{\mathcal{W}}^{CLIP}$ | ViT-B/32 | ✓ | 17.6 |
|  | ViT-B/16 | ✓ | 24.8 |
|  | ViT-L/14 | ✓ | 25.1 |
| ULTra$_{\mathcal{S}}^{CLIP}$ | ViT-B/32 | ✗ | 17.1 |
|  | ViT-B/16 | ✗ | 24.2 |
|  | ViT-L/14 | ✗ | 24.2 |
| ULTra$_{\mathcal{E}}^{CLIP}$ | ViT-B/32 | ✗ | 20.4 |
|  | ViT-B/16 | ✗ | 20.9 |
|  | ViT-L/14 | ✗ | 23.0 |
| ULTra$_{\mathcal{W}}^{Dino}$ | ViT-S/16 | ✓ | 25.8 |
|  | ViT-B/16 | ✓ | **26.5** |
| ULTra$_{\mathcal{S}}^{Dino}$ | ViT-S/16 | ✗ | 24.2 |
|  | ViT-B/16 | ✗ | 25.7 |
| ULTra$_{\mathcal{E}}^{Dino}$ | ViT-S/16 | ✗ | 22.9 |
|  | ViT-B/16 | ✗ | 23.0 |

Table 3: Comparison of different unsupervised segmentation methods on the Cityscapes dataset.

| Method | Model | Training | U. ACC |
|---|---|---|---|
| IIC | R18+FPN | ✓ | 65.1 |
| DINO | ViT-S/8 | ✓ | 71.3 |
| STEGO | ViT-S/8 | ✓ | 77.0 |
| DepthG | ViT-S/8 | ✓ | 80.4 |
| ULTra$_{\mathcal{W}}^{CLIP}$ | ViT-B/32 | ✓ | 78.7 |
|  | ViT-B/16 | ✓ | 80.9 |
|  | ViT-L/14 | ✓ | 82.4 |
| ULTra$_{\mathcal{S}}^{CLIP}$ | ViT-B/32 | ✗ | 78.3 |
|  | ViT-B/16 | ✗ | 80.9 |
|  | ViT-L/14 | ✗ | **82.8** |
| ULTra$_{\mathcal{E}}^{CLIP}$ | ViT-B/32 | ✗ | 78.1 |
|  | ViT-B/16 | ✗ | 70.4 |
|  | ViT-L/14 | ✗ | 75.5 |
| ULTra$_{\mathcal{W}}^{Dino}$ | ViT-S/16 | ✓ | 79.0 |
|  | ViT-B/16 | ✓ | 80.7 |
| ULTra$_{\mathcal{S}}^{Dino}$ | ViT-S/16 | ✗ | 77.4 |
|  | ViT-B/16 | ✗ | 80.8 |
| ULTra$_{\mathcal{E}}^{Dino}$ | ViT-S/16 | ✗ | 75.7 |
|  | ViT-B/16 | ✗ | 76.8 |

Table 4: Comparison of different unsupervised segmentation methods on the Potsdam dataset. The Training column Indicates if fine-tuning is required

achieves state-of-the-art results on several benchmarks.

For some models, such as ViT-L/14, no existing baseline is available for direct comparison, highlighting the versatility of `ULTra` across different architectures. Furthermore, we conducted an ablation study related to the model depth in Appendix A.

We hypothesize that ULTra's strong training-free performance arises from two key factors: (i) Comprehensive token utilization: leveraging information from *all* latent tokens yields richer representations than relying solely on the final `CLS` token; and (ii) Information-preserving aggregation: projecting embeddings into the input space *before* aggregation reduces information loss compared to methods that aggregate first. As shown in Figure 4, the `CLS` token loses substantial information that remains accessible through individual tokens.

Despite being zero-shot, our method is computationally intensive for semantic segmentation, since generating explanation maps requires computing multiple gradients per token (see Appendix B). Future work could explore approximation or more efficient gradient strategies (see Section 6).

### 5.3 Interpretability Evaluations

**Perturbation Test.** To assess the reliability of the explanation maps, we conduct a perturbation test by selectively altering image regions based on the explanation map $S_i^{(l)}$ for each token $i$. The perturbation is applied at the patch level while ensuring the total perturbed area remains consistent across all cases.

We consider two types of perturbations:

*(i) Positive Perturbation:* Removing highly relevant regions, which should significantly affect the token's representation.

*(ii) Negative Perturbation:* Removing less relevant regions, which should have minimal impact.

Two types of modifications are introduced. In the masking perturbation, selected patches are replaced with zeros, effectively removing the corresponding visual information. This is applied to both highly relevant

| Model | Perturb. | VOC | | Potsdam | | COCO | | Cityscapes | |
|---|---|---|---|---|---|---|---|---|---|
| | | Neg. ↓ | Pos. ↑ | Neg. ↓ | Pos. ↑ | Neg. ↓ | Pos. ↑ | Neg. ↓ | Pos. ↑ |
| ViT-B/32 | Mask | 13.16 | 15.51 | 12.35 | 14.02 | 12.62 | 15.21 | 9.95 | 14.42 |
| | Noise | 6.97 | 10.23 | 9.39 | 11.44 | 6.83 | 10.26 | 5.99 | 11.13 |
| ViT-B/16 | Mask | 14.68 | 17.5 | 13.91 | 15.51 | 14.16 | 17.39 | 11.33 | 16.27 |
| | Noise | 9.87 | 14.03 | 13.5 | 14.45 | 10.14 | 14.12 | 9.46 | 14.22 |
| ViT-L/14 | Mask | 6.66 | 9.43 | 6.61 | 8.99 | 6.38 | 9.31 | 5.36 | 8.86 |
| | Noise | 4.45 | 7.88 | 4.95 | 7.96 | 4.27 | 7.74 | 3.89 | 7.77 |

Table 5: Average token vector differences for ViT models under mask and noise perturbation tests across multiple datasets, highlighting the impact of positive and negative perturbations based on the relevancy map on token representations.

regions (positive masking) and less relevant areas (negative masking). In contrast, the noise perturbation introduces Gaussian noise to the same sets of patches, adding controlled randomness to test the robustness of token representations.

The noise follows a standard normal distribution with a standard deviation of 0.3. Figure 5 provides a detailed visualization of the perturbation test conducted on a sample image from the PASCAL VOC dataset using the CLIP ViT-B/32 model. It illustrates the model's explanation map (d), and compares the effects of both positive (b, c) and negative (e, f) perturbations applied through masking and Gaussian noise, demonstrating how altering semantically relevant regions leads to more significant changes in the model's internal representations.

To ensure fairness, the initial token's patch remains unchanged in both perturbations, as its information is directly propagated to the target token through skip connections. Given a perturbed representation $\tilde{z}_i^{(l)}$, we measure the deviation from the original token representation $z_i^{(l)}$ using the Euclidean distance, we then compute the average deviation over the entire dataset by selecting $k$ random tokens per image and aggregating the distances:

$$\bar{d}_{\text{Euc}} = \frac{1}{M} \sum_{j=1}^{M} \left( \frac{1}{k} \sum_{i=1}^{k} \|z_{i,j}^{(l)} - \tilde{z}_{i,j}^{(l)}\|_2 \right),$$

(14)

where $M$ is the number of images, and $k$ tokens are randomly selected from each. For our results, we set $k = 10$. The perturbation test results are presented in Table 5

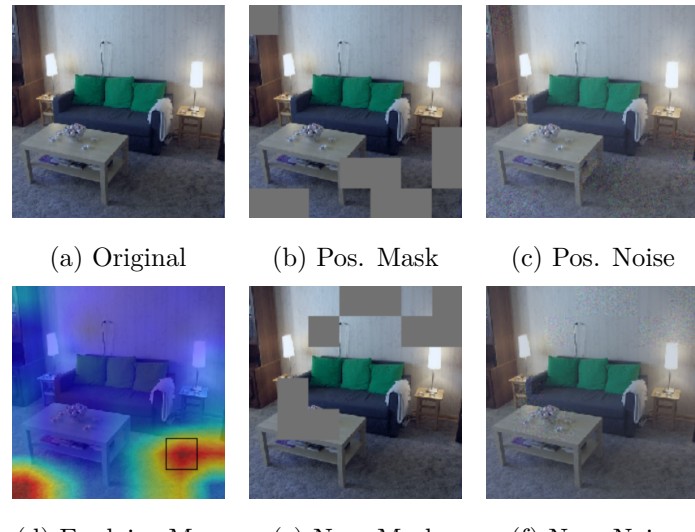

(a) Original    (b) Pos. Mask    (c) Pos. Noise

(d) Explain. Map    (e) Neg. Mask    (f) Neg. Noise

Figure 5: Effect of perturbations on a sample image from the PASCAL VOC dataset, where the model used is CLIP ViT-B/32.

Positive perturbations result in greater deviations compared to negative ones, confirming that the explanation maps effectively highlight influential regions. The consistency of these results across multiple datasets further validates the reliability of our interpretability framework. Similar results also hold for cosine similarity. However, due to redundancy and the fact that cosine similarity is bounded between 0 and 1, which results in weaker contrasts, we do not report these results.

How do I [ 2 0 M ] stop feeling bad about myself for having no relationship experience at all ? POST : It just seems like everyone I know has at least had a " thing " with someone by this point . I ' ve made out with a girl once ( who later told me that was a mistake ) and I feel like girls always reject me or only see me as a friend . Which is perfectly acceptable , but I ' m starting to get ups et that I ' ve never had any kind of relationship . I just got rejected by a girl who I thought was into me and I ' ve been feeling bad ever since . I just don ' t know what ' s wrong with me . I guess I ' m a little bit skin ny ( I work out regularly though ) , but I show er every day , dress pretty well , all that stuff .

I need help about those feelings POST : I am a 1 8 M , she ' s a 1 7 F . We ' ve got a troubles ome relationship which started as a pure friendship one year ago . I ' ve made mist akers , she made hers too . O ur last situation scenario is explained in here : Now I feel like I hate her , I used to adm ire her a lot , but I ' m really disappoint ed with her and with her character . But I just realized I still like her . So , well , yeah , I like her and hate her . And just after that bad situation happened I realized she also had that feeling . Well , now we both hate and love each other . What to do ? What to think ? What to feel ? add itional info : today our friend asked me for help with some calculations and I made a jo ke about our physics teacher . She laughed and smiled at me just like one year ago , but after she realized that , she seemed kind a [ gr ouch y ](

(a) TL;DR: I've had very bad luck with girls my whole life and I don't know how to get my confidence up.

(b) TL;DR: I still like her but my rational side says "no, she is a trash person".

Figure 6: Visualization of Token Contribution Scores $(\lambda_i^{(l)})$ highlighting the relevance of context tokens in interpreting the summary. Each token is colored proportionally to its $\lambda_i^{(l)}$ value. These visualizations demonstrate the model's ability to identify key semantic elements in the context for generating relevant summaries. Further analysis and examples are provided in Appendix H.

**Object Selection.** To quantify alignment, we compute the IoU by converting the explanation map $S_i^{(l)}$ into a binary mask $M_i^{(l)}$ and comparing it with the ground-truth mask. We propose the *Initial Token IoU (ITIoU)* metric, which measures how well the explanation maps of input tokens align with their respective class masks. The *ITIoU* is calculated as:

$$ITIoU^{(l)}(X) = \frac{1}{C} \sum_{i=1}^{C} \frac{1}{|\mathcal{T}_i|} \sum_{\mathbf{x}_j \in \mathcal{T}_i} \text{IoU}(M_j^{(l)}, G_i), \tag{15}$$

where $C$ denotes the number of classes, $\mathcal{T}_i$ represents the set of tokens associated with class $i$, $M_j^{(l)}$ is the binary segmentation mask for token $\mathbf{x}_j$ within class $i$, and $G_i$ is the ground-truth mask for class $i$ in image $\mathbf{x}$. The inner sum averages the IoU for tokens in $\mathcal{T}_i$ for each class, and the outer sum then averages across all classes. Using a threshold of 0.2, our *ITIoU* metric achieves an average score of 37.84% on the COCO-Stuff validation dataset and 39.51% on the PASCAL VOC dataset. A more detailed analysis of *ITIoU* is provided in Appendix F.

## 5.4 Interpretable Text Summarization

In this experiment, we used a Supervised Fine-Tuned (SFT) version of Llama-2-7B trained on the Ul-traFeedback Binarized (UFB) dataset (Cui et al., 2024). Additionally, we aligned the model to the text summarization task on the TL;DR dataset (Stiennon et al., 2022) using the Direct Preference Optimization (DPO) method (Rafailov et al., 2024) for 1,000 iterations, with a learning rate of $5 \times 10^{-6}$ and $\beta = 0.5$. To validate our framework, we selected the *preferred* response (TL;DR) of each sample in the dataset, denoted by $y$, and used it as the summary of the context $x$. The result can be seen in Figure 6

In example (a), semantically significant words such as 'relationship', 'experience', 'rejection', and 'never' are prominently highlighted, reflecting the model's interpretation of the person's struggles with relationships and feelings of rejection. Additionally, the highlighting of the question at the beginning of the context 'How do I stop feeling bad...' suggests the model recognizes the presence of uncertainty and a request for guidance, which is encapsulated in the summary as 'I don't know.'

In example (b), $\lambda_i^{(l)}$ scores reveal the model's focus on words such as 'feelings', 'hate', 'disappoint', 'love', and 'like', which correspond to the person's mixed emotions toward their girlfriend, as described in the summary. The apparent contradiction between 'love' and 'trashness' in the summary appears to be derived from these highlighted terms, suggesting the model understands the conflicting emotions present in the text. Furthermore, the focus on 'character' reflects the summary's judgmental tone, suggesting the model associates this term with a personality assessment.

**Quantitative Validations.** While the qualitative visualizations in Figure 6 illustrate ULTra's ability to highlight semantically relevant context tokens, we further assess the faithfulness of these attributions

using the *Comprehensiveness* metric (DeYoung et al., 2020). This metric measures how much the model's confidence in producing the gold summary decreases when important context tokens are removed.

Following Eq. 13, we compute token-level contribution scores $\lambda_i^{(l)}$ and select the top-$k$% tokens as the rationale set $R_k$. Using the pretrained language model $\pi$, we define the summary support score as the log-probability of the gold summary $y$ given context $x$:

$$S(x, y) = \log \pi(y \mid x) = \sum_{i=1}^{|y|} \log \pi(y_i \mid x, y_{<i}), \tag{16}$$

and compute the *Comprehensiveness* score:

$$\text{Comp}_k(x, y) = \frac{S(x, y) - S(x \setminus R_k, y)}{S(x, y)}, \tag{17}$$

where larger values indicate that the removed tokens were more necessary for maintaining model confidence.

As shown in Figure 7, ULTra consistently achieves higher $\text{Comp}_k$ values than the random baseline, which selects rationale sets $R_k$ uniformly at random. At small budgets ($k \leq 15\%$), the performance of both methods is similar; however, as $k$ increases, ULTra exhibits steady improvements, highlighting its ability to identify context regions that causally influence model predictions.

This quantitative analysis further confirms that ULTra's highlighted tokens are not only interpretable but also faithfully reflect the model's internal reasoning process. We also present additional qualitative examples and a detailed discussion of the metric in Appendix H.

These language-based experiments are not intended as a benchmark for interpretable summarization, especially given the absence of standardized evaluation metrics, but rather to demonstrate the versatility of ULTra across modalities. These experiments also highlight an important direction for future research: applying post-hoc interpretability techniques to better understand and align large language models within frameworks such as RLHF (Christiano et al., 2023; Stiennon et al., 2022; Ouyang et al., 2022) and direct preference optimization (Rafailov et al., 2024; Azar et al., 2023; Ethayarajh et al., 2024), where understanding model behavior and intent is critical.

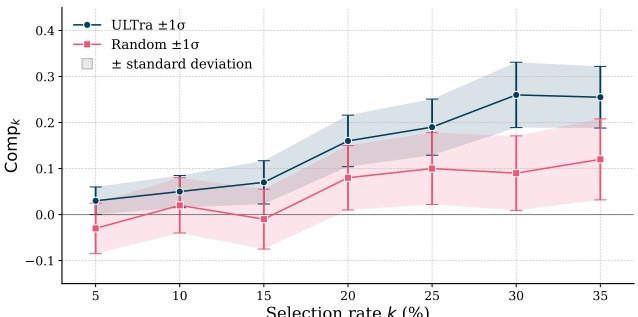

Figure 7: Average Comprehensiveness score $\text{Comp}_k$ on the validation set across different selection rates $k$ for ULTra and random baseline. ULTra shows a consistent increase, indicating that its selected tokens are critical for preserving model confidence in the summary.

## 6 Concluding Remarks and Limitations

*Summary.* We introduced a framework for interpreting latent tokens in Transformers, providing new insights into the semantic information encoded within them. Our method achieves state-of-the-art performance in unsupervised semantic segmentation across multiple datasets and settings, notably without requiring any additional training. Beyond segmentation, we validated the approach through perturbation tests and object selection, highlighting its broader applicability for probing Transformer behavior at the layer level.

*Discussion.* A central challenge in interpretability is its evaluation. Traditional methods often rely on class logits as the primary signal, employing strategies such as supervised semantic segmentation or perturbation tests to measure changes in these logits. However, even for explanations of final predictions, it remains unclear whether such approaches faithfully reflect the quality of interpretability. In our setting, many of these metrics are not directly applicable due to fundamental methodological differences. To address this, we evaluated our framework through the lens of unsupervised semantic segmentation. While baseline methods were

specifically designed for this task, our approach achieved superior results by aggregating explanation maps, demonstrating both the potential of latent token interpretability and the importance of effective aggregation strategies. For instance, clustering-based techniques could further improve performance, especially at smaller patch sizes; when the patch size is 8, the large number of tokens makes aggregation particularly challenging.

*Limitations and Future Work.* Despite operating in a zero-shot setting, our method incurs notable computational cost for semantic segmentation. High inference times result from computing multiple gradients for each token when generating explanation maps. We provide a FLOPs and estimated-runtime analysis (batch size 64) in Appendix B. Future work could alleviate this limitation by exploring more efficient strategies, including: (i) approximations in the attention mechanism (e.g., sparse attention or token pruning), (ii) gradient approximations (e.g., using fewer layers), and (iii) optimized computations such as parallel processing or alternative interpretability signals. In particular, we examines layer selection effect in Appendix A, where for some models, using middle layers instead of the last layer improves both accuracy and computational efficiency, whereas in others, a trade-off between performance and computational cost remains.

## 7 Broader Impact Statement

Our work contributes to the interpretability of large language models (LLMs) by identifying and analyzing latent token structures. While interpretability can enhance transparency, accountability, and trust in LLM-based systems, it also introduces potential risks. In particular, deeper insights into model internals could be misused for adversarial purposes, such as crafting more effective jailbreak prompts, reverse engineering proprietary models, or developing targeted manipulations that reduce model safety. Furthermore, interpretability methods risk conveying a false sense of security if stakeholders overestimate the completeness or reliability of the provided explanations.

Despite these risks, we believe that developing interpretability methods is ultimately beneficial for safe and responsible deployment of LLMs. By making the limitations and inner workings of these systems more visible, our work can help practitioners identify failure modes, mitigate biases, and design more robust safeguards. Careful communication of the scope and limitations of our method will be essential to minimize potential misuse and prevent overconfidence in the explanations it provides.

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

## Appendix

## A  Effect of Layer Depth in ViT Token Understanding

In this section, we analyze the impact of depth on our model's interpretability and segmentation performance, providing insights into the contribution of each layer. For smaller models such as ViT-B/32 in Figure 9, deeper layers generally carry more semantic significance. However, the contribution diminishes in the final layers, suggesting that a depth of around 13 layers might be more than sufficient for the ViT to effectively comprehend image content. This finding implies that even fewer layers might achieve comparable results, potentially reducing computational costs without compromising performance.

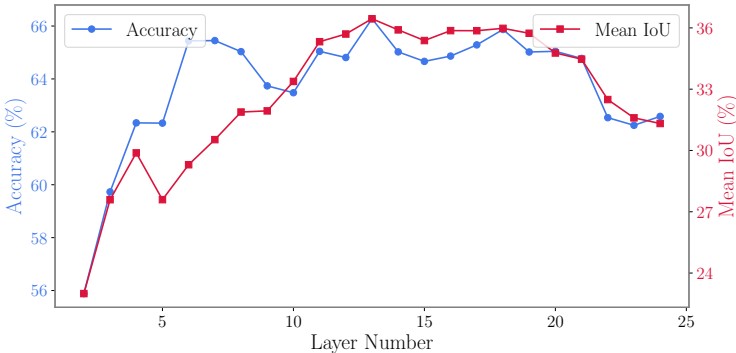

Figure 8: The segmentation performance of the CLIP ViT-L/14 model in an unsupervised setting, measured by accuracy and IoU, changes across different layers. As layer depth increases, both metrics improve at first, reaching their highest point in the mid-depth layers. However, performance slightly declines in deeper layers. This suggests that while deeper layers capture richer semantic details, they also introduce complexity that does not always improve segmentation. Additionally, the limitations of labeled datasets, especially the ambiguity in object definitions, further restrict the model's ability to achieve better segmentation, despite its strong interpretability.

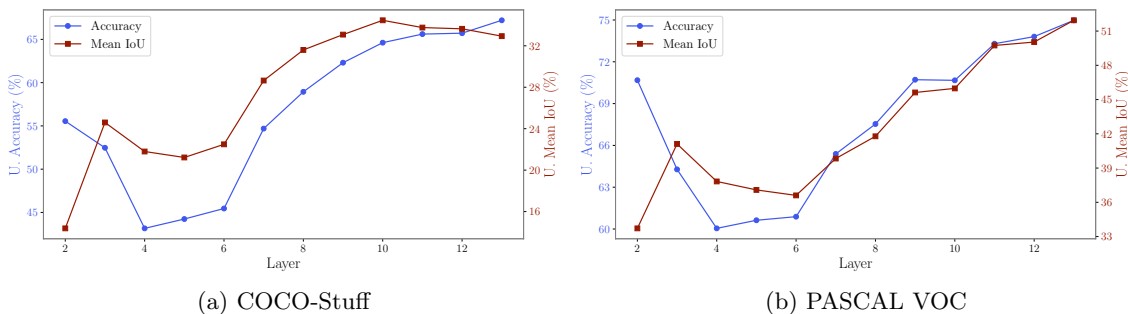

(a) COCO-Stuff       (b) PASCAL VOC

Figure 9: Ablation study on two evaluation metrics across layers ViT-B/32. These plots demonstrate a progressive improvement in semantic segmentation performance in the deeper layers of the transformer model. This enhancement is attributed to latent tokens capturing more meaningful segment structures, resulting in increasingly accurate and refined semantic representations.

We observe an intriguing behavior in the initial layers, where performance initially declines before improving. This phenomenon is also visually evident in Figure 4, where the attention maps in the first layer appear to focus on the entire image. This suggests that, initially, the token examines the image as a whole before selectively gathering information from tokens with similar characteristics.

In the CLIP ViT-L/14 model, deeper layers capture more refined and detailed feature representations. Unlike shallower models, where segmentation performance stabilizes early, ViT-L/14 benefits from its depth by gradually extracting richer hierarchical features. As shown in Figure 8, accuracy and Mean IoU improve as layers deepen, with segmentation performance peaking around the mid-depth layers. However, in deeper layers (beyond layer 20), segmentation quality slightly declines. This suggests that while the model gains a better understanding of high-level semantics, it loses some spatial precision. This trade-off occurs because later layers prioritize semantic abstraction over detailed spatial structures.

While deeper models like ViT-L/14 offer improved feature extraction, more layers do not always lead to better segmentation. Instead, an optimal balance between depth and spatial representation is necessary for effective segmentation.

We note that, for a fixed model, the overall performance remains relatively consistent across different datasets, as illustrated in Figure 4. This observation suggests the presence of a sweet spot—a region where the model achieves optimal performance.

A similar pattern can be observed in Figure 8, where a flat region in the curve represents this stable performance zone. Interestingly, the location of this sweet spot tends to be similar across datasets for the same model. In practice, it can be identified experimentally by evaluating the model's performance on a small validation subset and then verifying its consistency on other datasets.

## B    Complexity Analysis of ULTra for Segmenetation

In this section, we analyze the time complexity of `ULTra` for the segmentation task. To compute the final segmentation mask, our method requires the heatmaps of all tokens from a fixed layer. As a result, the computational complexity is significantly higher than that of methods that rely solely on a forward pass and are explicitly designed for segmentation. In our case, We perform a single forward pass and then $n$ backward passes (one backward pass per token, where $n$ denotes the number of tokens). The backward passes constitute the primary source of computational overhead.

To quantify this cost, we report the total number of FLOPs needed for both the forward and backward passes at a fixed layer. The results are presented in Table 6. I-t is important to note that, although our segmentation approach is computationally intensive, ULTra is primarily designed as an interpretability framework. Segmentation is just one application that benefits from the generated heatmaps. While the framework was not specifically developed for segmentation, it nonetheless achieves state-of-the-art performance in unsupervised semantic segmentation and notably, without any training. Moreover, segmentation is also included as part of our evaluation.

Figure 10: Visual comparison of segmentation quality between $\text{ULTra}_\mathcal{S}$ and $\text{ULTra}_\mathcal{W}$ on several examples from the COCO-Stuff (CLIP ViT-B/32). Each row corresponds to a different example, and the columns show the original image, the heatmap generated by $\text{ULTra}_\mathcal{S}$, and the refined heatmap produced by $\text{ULTra}_\mathcal{W}$.

## C    Qualitative Analysis of $f_w$

In this section, we visually present the improvement of $\text{ULTra}_\mathcal{W}$ over $\text{ULTra}_\mathcal{S}$ when $\text{ULTra}_\mathcal{S}$ is used as the initial segmentation. Figure 10 provides several examples illustrating how this improvement is captured during the learning of $f_w$. Recall that $\text{ULTra}_\mathcal{W}$ is trained by perturbing the negative regions in the image and learning a representation that is robust to such perturbations. This process encourages the model to develop features that are less sensitive to the negative regions and more focused on the positive regions. Consequently, the resulting heatmaps become more centered on the target object that was previously captured by $\text{ULTra}_\mathcal{S}$.

| Model | Forward (FLOPs) | Layer 3 | Layer 5 | Layer 7 | Layer 9 | Layer 11 | Layer 13 |
|---|---|---|---|---|---|---|---|
| ViT-B/32 | $5.7 \times 10^{11}$ | $5.6 \times 10^{12}$ | $2.0 \times 10^{13}$ | $4.4 \times 10^{13}$ | $7.7 \times 10^{13}$ | $1.1 \times 10^{14}$ | $1.7 \times 10^{14}$ |
| Ratio | $1\times$ | $10\times$ | $35\times$ | $77\times$ | $135\times$ | $193\times$ | $298\times$ |
| ViT-B/16 | $2.3 \times 10^{12}$ | $9.2 \times 10^{13}$ | $3.3 \times 10^{14}$ | $7.3 \times 10^{14}$ | $1.2 \times 10^{15}$ | $1.9 \times 10^{15}$ | $2.8 \times 10^{15}$ |
| Ratio | $1\times$ | $40\times$ | $143\times$ | $317\times$ | $521\times$ | $826\times$ | $1217\times$ |

Table 6: FLOPs required to compute a forward pass and to generate gradient-based explanation maps targeting different layers for batch size of 64. Ratios show the multiple relative to one forward pass.

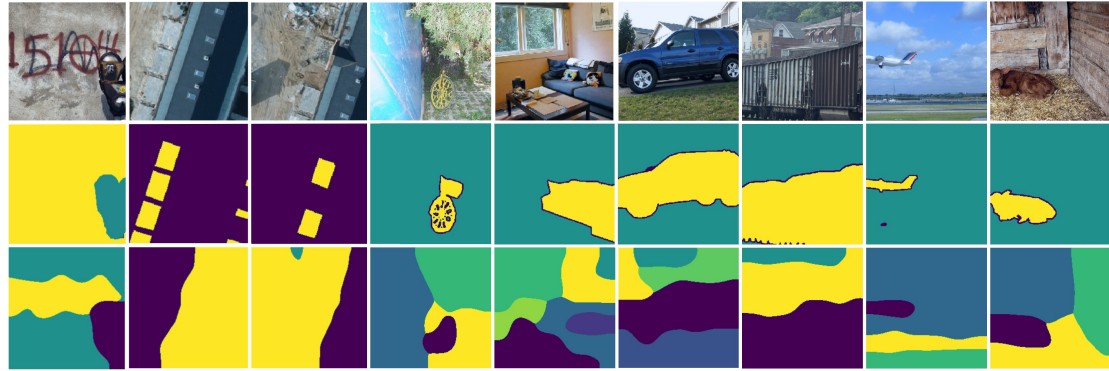

Figure 11: This figure illustrates challenging samples from the COCO-Stuff 27, Potsdam-3, and PASCAL VOC that contribute to low performance due to poor dataset labeling quality. The first row displays the original images, the second row shows the ground truth labels and the third row presents our predicted segmentation.

It is important to note that though $\text{ULTra}_{\mathcal{W}}$ depends on the initial segmentation, we do not expect it to perform worse than the initial results. This is because the target token itself always lies within the positive region, and surrounding areas with positive values tend to be reinforced. In essence, $\text{ULTra}_{\mathcal{W}}$ sharpens the heatmap around these positive regions. As shown in Figure 10 $\text{ULTra}_{\mathcal{W}}$ further captures the regions identified by $\text{ULTra}_{\mathcal{S}}$.

## D  Datasets Limitations

A significant challenge in deep learning is obtaining labeled datasets, which are essential for the success of deep learning methods. However, labeling can be ambiguous, as objects or attributes may be labeled separately or combined as a single entity. Our method demonstrates how ViT interprets images in a zero-shot setting, capturing fine-grained or general representations. Notably, our method's predictions often exhibit logical consistency that surpasses the ground truth, effectively identifying relationships between objects.

However, since our approach does not involve supervised training, it cannot adapt to detect only the specific objects labeled in the dataset. As a result, while the logical quality of the predictions is high, the numerical metrics may decline due to mismatches with the dataset's ground truth labels. Figure 11 illustrates this phenomenon with examples, showing instances where our method successfully captures unlabeled objects that are omitted in the dataset annotations.

## E  Datasets & Models

### E.1  Datasets

We utilize a combination of datasets to provide a diverse testing ground for evaluating our method across both standard and challenging perspectives in semantic segmentation and interpretability evaluations.

- **COCO-Stuff 27** Lin et al. (2014): A subset of the COCO dataset, featuring complex real-world scenes with pixel-level annotations across various object categories.

- **PASCAL VOC 2012** Everingham & Winn (2011): A widely used benchmark containing pixel-level annotations for foreground objects in structured scenes.

- **Potsdam-3**: A high-resolution aerial-view dataset capturing urban landscapes, including buildings, roads, and vegetation, presenting additional challenges due to its large-scale top-down perspective.

- **Cityscapes** Cordts et al. (2016): An urban street scene dataset with fine-grained pixel-level annotations, enabling the evaluation of segmentation performance in structured environments.

These datasets allow for a comprehensive evaluation of our approach across different environments, ensuring robustness across diverse segmentation challenges.

### E.2 Models

We employ various **Vision Transformers (ViTs)** pre-trained on large-scale datasets. These models process images as non-overlapping patches and employ self-attention mechanisms across multiple layers to capture long-range dependencies.

**Transformer Architectures**

**CLIP ViT** Radford et al. (2021): A vision transformer trained using contrastive learning on 400 million image-text pairs. It encodes images into a shared embedding space with text prompts. CLIP variants include:

- **ViT-B/16**: Consists of 12 transformer layers, a hidden size of 768, and processes images with 16×16 patch resolution.

- **ViT-B/32**: Similar to ViT-B/16 but with a 32×32 patch resolution, reducing computational cost at the expense of finer details.

- **ViT-L/14**: A larger model with 24 transformer layers, a hidden size of 1024, and a 14×14 patch resolution, providing enhanced feature extraction.

**DINO ViT** Caron et al. (2021): A self-supervised vision transformer trained using knowledge distillation without labeled data. It learns image representations by maximizing similarity between different augmented views. Evaluated variants:

- **ViT-S/16**: A smaller model with 12 transformer layers, a hidden size of 384, and a patch size of 16×16.

- **ViT-B/16**: A larger model with 12 layers, a hidden size of 768, and a patch size of 16×16, providing stronger feature representation.

## F   ITIoU Analysis

This section evaluates the effectiveness of *ITIoU* in assessing the performance of our object selection process. As expected, the final layers exhibit superior performance compared to the earlier layers, consistent with the results illustrated in Figure 13. This improvement highlights the increasing relevance of features in deeper layers for accurate object selection. Additionally, we observe the existence of an optimal threshold, $\tau$, which significantly influences segmentation performance. This phenomenon is depicted in Figure 12, where performance trends are analyzed across different threshold values.

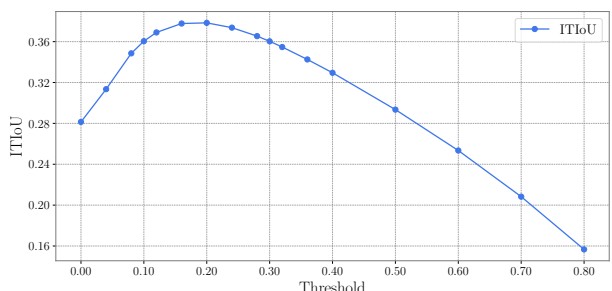

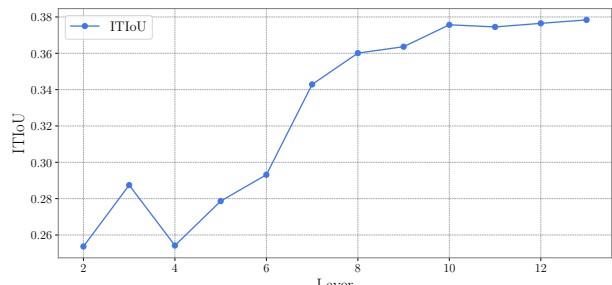

Figure 12: Impact of varying the threshold $\tau$ on *ITIoU* performance for COCO-Stuff 27 dataset. The plot demonstrates the existence of an optimal $\tau$, where segmentation performance is maximized.

Figure 13: Layer-wise *ITIoU* analysis for COCO-Stuff 27 dataset. Final layers perform significantly better than earlier ones due to their ability to capture high-level semantic features. This progression is evident in the increasing *ITIoU* values.

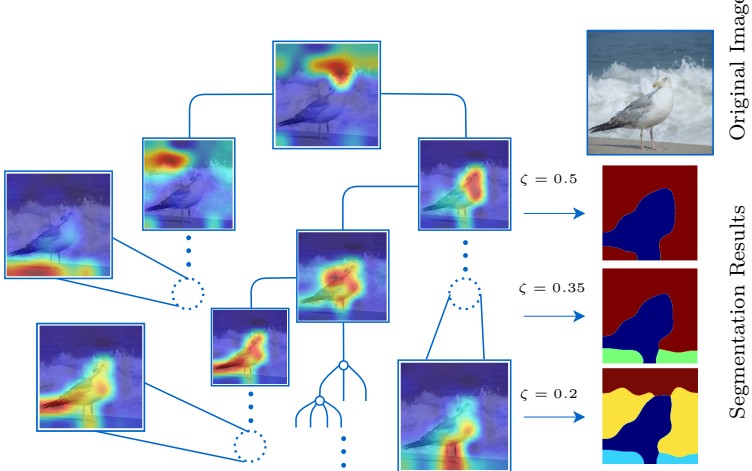

Figure 14: Hierarchical clustering tree showing the grouping of token explanation maps for all tokens in a latent layer of the Vision Transformer, not limited to the `CLS` token. Each leaf node represents a single token explanation map, while higher-level nodes show aggregated clusters based on a clustering threshold $\zeta$, which controls the level of detail. Lower $\zeta$ values reveal finer details, while higher values create broader, more general clusters.

## G Threshold-Based Segmentation

Hierarchical clustering is used to segment the token explanation maps, where the threshold parameter $\zeta$ controls the level of granularity. Lower values of $\zeta$ produce fine-grained segmentations, while higher values merge similar clusters, leading to broader groupings.

Figure 14 illustrates the hierarchical clustering tree, where each leaf node represents a single token explanation map. As $\zeta$ increases, multiple explanation maps are grouped into larger clusters, reducing segmentation granularity. The rightmost part of the figure shows how different $\zeta$ values affect the final segmentation output.

Figure 15 further quantifies the impact of $\zeta$ on segmentation performance for the COCO-Stuff 27 dataset. Accuracy and Mean IoU metrics are plotted as a function of $\zeta$, demonstrating that while different values of $\zeta$ yield varying levels of segmentation detail, our method remains robust across a range of threshold values.

The results confirm that selecting an appropriate $\zeta$ enables the model to segment objects at different levels of abstraction, demonstrating the adaptability of pre-trained Vision Transformers in unsupervised semantic

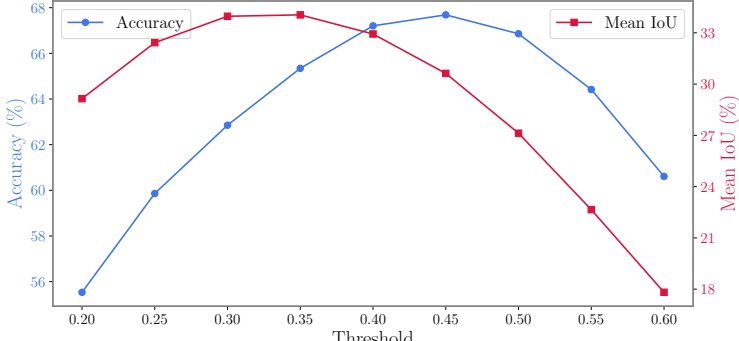

Figure 15: Impact of the clustering threshold $\zeta$ on segmentation performance for the COCO-Stuff 27 dataset. Accuracy and Mean IoU metrics show how different values of $\zeta$ affect segmentation quality.

| Dataset | U. ACC | U. mIoU |
|---|---|---|
| COCO-Stuff 27 | 67.2 | 32.9 |
| PASCAL VOC | - | 51.9 |
| Potsdam | 74.6 | - |

Table 7: `ULTra` results across different datasets using the same threshold $\zeta = 0.4$. Accuracy (U. ACC) and Mean IoU (U. mIoU) are reported where available.

segmentation. Table 7 summarizes the segmentation performance of our method across multiple datasets using a fixed threshold $\zeta = 0.4$. However, as discussed in Section D, not all threshold values may be equally suited for standard datasets.

## H  Further Analysis of Text Summarization

### H.1  Additional Examples

To complement the qualitative results presented in Section 5.4, we provide additional visualization examples of the TL;DR summarization task in Figure 16. These samples further validate the interpretability of the ULTra framework by demonstrating its ability to consistently highlight semantically salient tokens across diverse contexts. The extended examples illustrate how token-level relevance scores align with the distilled summaries.

**Explanations:**

(**a**): Semantically significant words such as *"care deeply," "depression," "cutting,"* and *"upset"* are prominently highlighted, reflecting the model's interpretation of the partner's struggles and the narrator's emotional reaction. The focus on *"love"* and *"don't know"* further shows the model capturing both the romantic attachment and uncertainty, which are condensed into the summary.

(**b**): The model highlights tokens like *"team," "emotional," "underdog win,"* and *"cry/teary,"* focusing on the parts about bonding with a team and the emotional reaction to underdog victories. This attention shows the model captures the link between shared struggles and the narrator's tears, which is distilled into the TL;DR.

(**c**): The model highlights tokens such as *"pretty girl," "father," "shy," "well with the dad,"* and *"unsure,"* which emphasize both the attraction and the social barriers around the situation. Strong attention is also given to the repeated question *"How should I proceed / how should I go about asking out,"* showing the model identifies the narrator's uncertainty as the key theme. By balancing the highlighted

references to family dynamics and the narrator's hesitation, the model distills the post into the TL;DR: asking the girl out without misunderstandings or awkwardness.

**(d)**: The model highlights tokens such as *"mid year evaluation," "pay," "amount of units," "24 units away from graduating,"* and *"working full time over the summer,"* which directly connect to the question of salary adjustment. Attention is also given to phrases about reflecting the *"current amount of units"* taken, showing the model identifies the link between academic progress and pay. These focused regions explain why the TL;DR condenses the post into the core concern of aligning mid-year pay with completed units. By contrast, the final sentence (*"I called HR... finals coming up"*) receives lower scores, as it conveys anxiety rather than information directly relevant to the pay issue.

**(e)**: Here, the model highlights tokens such as *"younger brother died," "suddenly," "impacting him heavily," "support him,"* and *"help him through this grief process,"* which directly connect to the main concern of coping with sudden loss. Broader attention is placed on sentences describing the death and its emotional toll, showing the model captures the central event and the narrator's intent to provide support.

**(f)**: The model highlights tokens such as *"pulled over," "red light," "ran through,"* and *"verbal warning,"* which directly capture the core sequence of events leading to the summary. Broader attention is given to the description of being stopped and told not to repeat the behavior, showing the model identifies this as the essential outcome.

### H.2 Quantitative Validations

To quantitatively assess the faithfulness of the highlighted tokens produced by ULTra, we adopt the *Comprehensiveness* metric introduced by DeYoung et al. (2020) for our task, measuring the impact of removing important context tokens on the model's behavior.

First, as in Eq. 13, we compute token-level contribution scores $\lambda_i^{(l)} \in \mathbb{R}^+$ denotes the contribution of the $i$-th context token to interpreting the summary $y$ at a fixed layer $l$.

Based on these scores, we select the top-$k\%$ of context tokens to form the rationale set $R_k$. We then evaluate how the removal of these tokens affects the model's ability to support the gold summary using the same pretrained model.

To evaluate the effect of removing important tokens, we define a scoring function $S(x, y)$ that quantifies how well the model supports the gold summary $y$ given the context $x$. Let the language model $\pi$ define the conditional distribution:

$$\pi(y \mid x) = \prod_{i=1}^{|y|} \pi(y_i \mid x, y_{<i}), \tag{18}$$

where $y_i$ denotes the $i$-th token of the summary and $y_{<i}$ the prefix tokens preceding it. We use the *log-probability* of the gold summary given the context as the score:

$$S(x, y) = \log \pi(y \mid x) = \sum_{i=1}^{|y|} \log \pi(y_i \mid x, y_{<i}). \tag{19}$$

Intuitively, $S(x, y)$ reflects the model's confidence in generating the reference summary when conditioned on the provided context. Let $x$ be the original context and $x \setminus R_k$ be the context with the selected tokens removed. The Comprehensiveness score at selection rate $k$ is then defined as:

$$\text{Comp}_k(x, y) = \frac{S(x, y) - S(x \setminus R_k, y)}{S(x, y)}, \tag{20}$$

where larger values indicate that removing the selected tokens leads to a larger drop in entailment, implying that these tokens are more *necessary* for preserving the semantic content of the summary.

: Is it fair to tell someone you can 't be with them unless they seek outside help for their issues ? POST : This account is just becoming Secret Time ... but anyways : I was wondering if anyone 's been in this sort of situation . I 'm seeing someone I care deeply about ( sure , the Love thing ) and they 're going through some rough times related to loss and just general depression ( cut ting , etc ). Recently , the cutting has started up again ( it had only happened for a short period of time before and I found out pretty soon afterwards /at the end of it / ended it ) and I don 't know . It makes me upset , and is draining , and I think it 's sort of evidently altering how I feel about them ' rom antically ' speaking , which I think in turn is just making it

(a) TL;DR: I'm seeing someone I care about get cut again and they're getting depressed, and I'm kind of in love with them and don't know what I should do.

of Reddit ! What makes you cry ? Sports Edition ! ( Story inside ) POST : So maybe it 's just me , maybe not . I 've never been an emotional kid growing up . I rarely cried in the usual tear jer ker movies or other emotional experiences , but ever since playing soccer all throughout high school and going through thick and thin with my team , It always gets me emotional to see the under dog win in just about any sport , and I get a little te ary . Is this normal or am I just the odd one out ? If you 've ever cried after playing or watching any sports event , whether it 's the Super b owl , the Wimbledon , or the World Table Tennis Championships , share your story please .

(b) TL;DR: I cry when I see an underdog win in just about any sports, does anyone else? Share your story!

[ 2 1 M ] like a pretty girl [ 1 8 - 2 2 F ], but she is always with her father . How should I proceed ? POST : I go to a certain shop occasionally , and the cashier is a really pretty girl who seems about my age ( 2 1 ). I don 't talk to her much . I think she 's shy because her dad ( the owner of the shop ) is always standing behind her . I get on well with the dad ( better than his daughter , who is very quiet in my presence ). I 'm quite sure he thinks well of me , and I think he knows I might like his daughter , but I 'm unsure . How should I go about asking out his daughter without offending him or making the daughter uncomfortable ? Even so , the daughter might not even like me in that way so it 's a tricky situation .

(c) TL;DR: How do I go about asking the girl out without any misunderstandings/awkwardness?

ceiving a raise as an intern POST : So my mid year evaluation is coming this Friday . The pay is directly correlated with the amount of units you have taken . When I was hired , I was a junior and my pay was set at 1 8 /hr and I gladly accepted . Now I am 2 4 units away from graduating and will be working full time over the summer . The average pay for a senior is 2 1 /hr . I was going to say if there was anyway my pay could reflect the current amount of units I have taken . I called HR and they said I should be fine , but I still need my bosses approval . I am getting a bit anxious and with finals coming up its not too great . Any suggestions ?

(d) TL;DR: How do I get my mid-year pay to reflect the increase in units I have taken.

[ 2 5 F ] with my SO [ 2 6 M ] of 4 years , how to help him cope with sudden loss . POST : This is a brief and easy one . Two days ago , my SO 's younger brother died rather brutally and ( ob viously ) suddenly . I have only met this man once in the 4 years we 've been together , but obviously this is impacting him heavily . He has been with family out of town for the last couple days and is on his way home to me ( and our daughter ) now . ( I had to stay home and continue work and take our daughter to school .) How do I best support him through this ? I will obviously be attending the service with him in a couple weeks . Im doing my best to make sure that the house is clean and life is as normal as possible when he gets home , so he has very little to stress about . But what can I do for him now ? How can I best help him through this grief process ?

(e) TL;DR: Boyfriend's brother died suddenly and it's impacting him very hard. What can I do to help him through this?

a police officer ever been a Good Guy Greg to you ? Describe it POST : I was once pulled over coming home from a friends Halloween party for running a red light . I went through right as it changed , but ran through it nonetheless . The officer pulled me over and did the usual deal , Licence and registration , do you know why i pulled you over ? I told him i knew that I had ran the light and my reasoning was that I was just dog tired , as it was 3 : 3 0 AM and I just wanted to get home . He asked me if there had been any drinking or drugs at the party and I hurried ly said no , as I was only 1 7 at the time . He went back to his cruiser and ran my licence and plates and then came back and informed me that i was going to get a verbal warning , i .e don 't do it again kind of thing , and then he escorted me home , which was about five miles away .

(f) TL;DR: I pulled over for running a red light, was told not to do it again.

Figure 16: Comparison of six different summarization examples. Each subfigure illustrates how the model interprets and condenses the input text, highlighting variation across samples.

In Figure 17, we compare $\text{Comp}_k$ of ULTra, i.e., selection is performed based on the ULTra method as defined in Eq. 13, with the same-sized random selection as a baseline, denoted as *Random*. We report the results across multiple selection rates, averaged over 300 samples randomly drawn from the validation set of the TL;DR dataset. The results demonstrate that ULTra consistently achieves higher $\text{Comp}_k$ values than the Random baseline, particularly as the selection budget increases. At lower selection rates, ULTra is only slightly better than Random, reflecting the difficulty of pinpointing influential tokens with very limited budgets. However, as more tokens are selected, ULTra identifies context regions that have a clearer causal impact on the model's output, while the Random baseline exhibits negative or unstable behavior, especially at smaller budgets.

**Limitations.** While the proposed Comprehensiveness metric offers an intuitive way to assess ULTra's effectiveness in identifying influential context tokens for summarization, it does not provide a complete evaluation of interpretability. This metric focuses only on the causal impact of token removal and does not

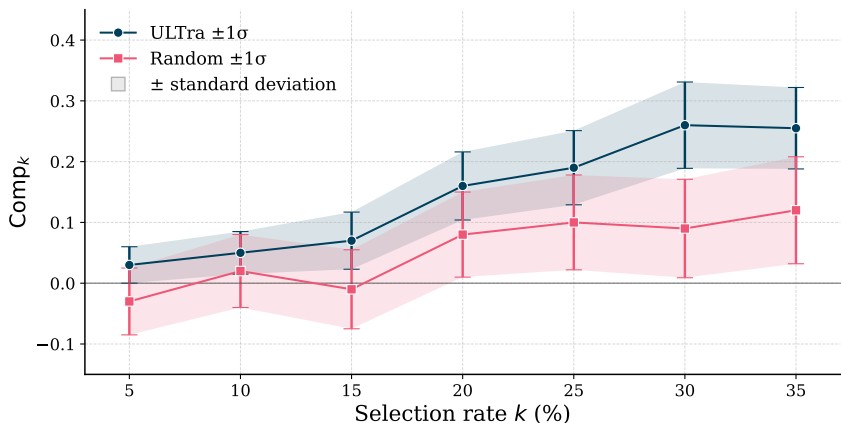

Figure 17: Average Comprehensiveness score $\text{Comp}_k$ on the validation set across different selection rates $k$ (%) for ULTra and a random baseline. At small budgets ($k \leq 15$), ULTra performs only marginally better than Random; as $k$ increases, it exhibits steadier gains, whereas Random shows negative values at low $k$ and non-monotonic behavior beyond 20%.

capture other important aspects such as sufficiency, stability under paraphrasing, or alignment with human rationales. It is also sensitive to the choice of language model and may reflect model-specific behaviors. A more comprehensive benchmark and evaluation strategy for interpretable text summarization should be developed in future work, combining multiple faithfulness measures, human evaluations, and robustness analyses.

# I  Transformer Architecture

The architecture of a typical Transformer can be formulated as follows: the input $X$ is split into $n$ tokens $\{\mathbf{x}_i\}_{i=1}^n$. After tokenization, token embeddings $\{\mathbf{e}_i\}_{i=0}^n$ are computed, where $\mathbf{e}_0$ corresponds to the CLS token. Positional encodings $\text{PE}_i$ are added to the $i$-th token embedding to incorporate spatial information, resulting in the latent token representation $\mathbf{z}_i^{(1)} = \mathbf{e}_i + \text{PE}_i$. Here, $\mathbf{z}_i^{(l)}$ represents a latent token, where $l$ denotes the layer index with $l \in \{1, \ldots, L\}$ and $L$ is the total number of layers in the Transformer, and $i$ represents the $i$-th token within the $l$-th layer.

For each head $h \in \{1, \ldots, H\}$ in the multi-head attention mechanism, the queries, keys, and values corresponding to the $i$-th token are obtained via linear transformations, projecting the latent token of dimension $d$ into dimension $k$. For $\forall l \in \{2, \ldots, L\}$ :

$$Q_h^{(l)}(\mathbf{z}_i^{(l-1)}) = (W_{h,q}^{(l)})^T \mathbf{z}_i^{(l-1)}$$

$$K_h^{(l)}(\mathbf{z}_i^{l-1}) = (W_{h,k}^{(l)})^T \mathbf{z}_i^{(l-1)}$$

$$V_h^{(l)}(\mathbf{z}_i^{(l-1)}) = (W_{h,v}^{(l)})^T \mathbf{z}_i^{(l-1)} \tag{21}$$

where $W_{h,q}^{(l)}, W_{h,k}^{(l)}, W_{h,v}^{(l)} \in \mathbb{R}^{d \times k}$. The attention weights for each token pair $(i, j)$ at layer $l$ and head $h$ are computed as:

$$\alpha_{h,i,j}^{(l)} = \text{softmax}_j \left( \frac{\langle Q_h^{(l)}(\mathbf{z}_i^{l-1}), K_h^{(l)}(\mathbf{z}_j^{(l-1)}) \rangle}{\sqrt{k}} \right). \tag{22}$$

The $i$-th token is then updated by summing over the weighted values across all heads:

$$\bar{\mathbf{u}}_i^{(l)} = \sum_{h=1}^H (W_{c,h}^{(l)})^T \sum_{j=1}^n \alpha_{h,i,j}^{(l)} V_h^{(l)}(\mathbf{z}_j^{(l-1)}), \tag{23}$$

where $W_{c,h} \in \mathbb{R}^{k \times d}$. The updated token representation $\mathbf{u}_i$ after the attention layer is computed as:

$$\mathbf{u}_i^{(l)} = \text{LayerNorm}(\mathbf{z}_i^{(l-1)} + \bar{\mathbf{u}}_i^{(l)}). \tag{24}$$

Each token then passes through a feed-forward network:

$$\bar{\mathbf{z}}_i^{(l)} = (W_2^{(l)})^T \text{ReLU}((W_1^{(l)})^T \mathbf{u}_i), \tag{25}$$

$$\mathbf{z}_i^{(l)} = \text{LayerNorm}(\mathbf{u}_i + \bar{\mathbf{z}}_i^{(l)}). \tag{26}$$

Here, $W_1^{(l)} \in \mathbb{R}^{d \times m}$, $W_2^{(l)} \in \mathbb{R}^{m \times d}$.

