# OpenReview forum: "ULTra: Unveiling Latent Token Interpretability in Transformer-Based Understanding and Segmentation"
_TMLR — Accepted by TMLR_

### Review · Reviewer_T9SA · 2025-08-20

**Summary Of Contributions:**

The paper proposes Ultra, a method designed to provide interpretability to tokens in Visual Transformer architectures. While the method was initially developed for general use, it focuses specifically on unsupervised semantic segmentation (USS). The method is evaluated on multiple datasets (Pascal VOC, COCO-Stuff, Cityscapes, Potsdam) and also reports results in text analysis.

Side note: The paper is stated to be of standard length (12 pages), meaning the appendices are not considered mandatory material. If they are intended to be treated as part of the paper, they should be moved into the main text.

**Additional Comments:**

Followin discussion with authors, I believe that if changes assumed in the comment are made, the paper moves in the positive area.

**Audience:**

Yes

**Audience Explanation:**

There two strengths:
- the unsupervised semantic segmentation, where the paper shows strong results.
- the interpretability of token in ViT

**Broader Impact Concerns:**

There are no concerns here.

**Claims And Evidence:**

No

**Claims Explanation:**

The paper does not make clear claims and does not present clear (accompanied by detailed explanation) results to round claims.

**Requested Changes:**

**Strengths**:

    - The paper is technically correct: I have checked the mathematics and found no major errors.
    - The results are strong: I have reviewed multiple papers on unsupervised semantic segmentation, and the results presented here are indeed competitive. The paper could stand on its own as a contribution to USS.

**Weaknesses**:

    - The paper is not particularly clear in its purpose. It oscillates between “interpretability of the token” and unsupervised semantic segmentation. From the abstract and introduction, I initially assumed the focus was on the former, yet the evaluation is centered on USS, with some addition on text. Several aspects remain unclear:

    - How is the success of “measuring the contribution of latent tokens” evaluated? Since the paper does not explicitly state its main objective, it is unclear: contribution of tokens to what?
   - The paper states that it measures the influence weight of a token in a specific representation (before Eq. (5)). This is fine, but why is it relevant? Why is it more important than simply aggregating the weights between the two?

   - What loss function is used for USS training? This is related to the previous point—what is expected from token interpretability? I believe the paper should start with a hand-drawn example illustrating what is expected.
   - The same issue applies to the text analysis section.
  -  The interpretability results are not compelling. The figures do not communicate much. We see heatmaps (saliency maps) in Figures 1, 3, 4, and 6, but it is unclear what we are supposed to observe and how this demonstrates successful token interpretability. For instance, in Figure 3, the token heatmap focuses on a dog, and then a segmentation mask around the dog is produced. However, the segmentation mask already conveys all the necessary information.

**Concluding weakness**: The paper emphasizes “unsupervised” settings, but interpretability in unsupervised contexts is problematic—since there are no ground truths, it is difficult to judge whether the results are good or not. As a result, the paper lacks a metric for evaluating token contribution or interpretability in representation. To me, the work currently feels like a work in progress rather than a finished piece.

**Minor issues**:

   - text : "experimetns"
    - Figure 2 comes in text after figure 3

---

> ### Author Response · Authors · 2025-10-08
> **additional concerns?**
>
> Given our previous discussion [here](https://openreview.net/forum?id=vL3pmJjGDQ&noteId=q1tUkJpMUD) and the updated version of the paper [here](https://openreview.net/forum?id=vL3pmJjGDQ&noteId=eKGrOK5cMt), could you please let us know if there are any additional concerns or points we should address?

---

> > ### Comment · Reviewer_T9SA · 2025-10-14
> > **Explanation is sufficient**
> >
> > Thank you for the detailed explanation and answer. From my perspective there is no additional question.

---

### Review · Reviewer_v9J2 · 2025-09-22

**Summary Of Contributions:**

The paper introduces ULTra, a framework to describe latent tokens in Transformer models in terms of the semantic contribution of individual tokens. Although designed as a general interpretability tool, its first use and result are towards unsupervised semantic segmentation(USS). The framework is applied to pre-trained models without requiring fine-tuning and is tested on multiple vision datasets, with an additional qualitative test on text summarization.

Strengths

1.	Leverages attention-based interpretability and logically extends it to latent token analysis is an interesting idea.

2.	The proposed method achieves state-of-the-art performance on standard unsupervised semantic segmentation benchmarks, showcasing the existence of rich semantic structure in latent representations without fine-tuning.

Weaknesses

1.	Vague Goal: The main goal of the paper is vague. It varies from being a method for latent token interpretability to being a method for unsupervised semantic segmentation. Both the abstract and the introduction place the work within interpretability, yet the experiments and results concentrate primarily on demonstrating USS performance. This is unclear in terms of the main contribution of the paper. There are a few things that remain unanswered:

(1) How do we measure the success of interpreting a latent token? The paper proposes a framework to measure token contribution, but we have no idea what this contribution is to, and how we can check whether it is correct or not.

(2) The paper introduces a function f(z_i^(l)) to yield a scalar output from a token embedding so that it can be backpropagated, yet the rationale for the specific choices (sum, energy, or weighted projection) is performance on downstream tasks, not fidelity to the token's "true" meaning.

(3) What is the metric for determining interpretability itself? The indirect and insufficient use of USS performance as a proxy for the quality of interpretation.

2.	Unpersuasive Interpretability Assessment. Qualitative results used to demonstrate interpretability are not quite convincing. The visualizations shown in Figures 2 and 4 show heatmaps corresponding to objects, but this is normal behavior that gives a good segmentation. The heatmaps do not reveal much new insight into the model's reasoning from what is already gained through the final segmentation mask. It does not seem clear what specific new insight is gained through these visualizations.

3.	The Unsupervised Problem. The work points out its "unsupervised" nature, but this actually makes interpretability assessment basically difficult. There is no ground truth in an unsupervised setup on what a latent concept should be. Without this, any statement on finding semantically significant patterns is human and can't be rigidly confirmed. The work does not mention this inherent limitation, thus making the interpretability claims appear unproven.

Minors: Typo in Section 3.2, "desiging" -> "designing".

**Additional Comments:**

None

**Audience:**

Yes

**Audience Explanation:**

Even if the interpretability evidence is not fully convincing, I believe several segments of TMLR’s audience would still find the paper’s findings and methodology interesting

**Claims And Evidence:**

Yes

**Claims Explanation:**

The paper convincingly supports its USS performance claims (solid zero-shot results on standard benchmarks; no fine-tuning) but does not provide clear, convincing, and falsifiable evidence for its interpretability claims.

**Requested Changes:**

Please refer to the weakness section.

---

> ### Author Response · Authors · 2025-10-08
> **Response to Reviewer v9J2**
>
> ### Authors’ Response to Reviewer v9J2
>
> We sincerely thank Reviewer v9J2 for recognizing that our work is both interesting in nature and strong in USS performance.
>
> The reviewer’s main concern relates to the **evaluation process**. We refer the reviewer to our earlier discussion with Reviewer T9SA ([link](https://openreview.net/forum?id=vL3pmJjGDQ&noteId=q1tUkJpMUD)) and the corresponding updates in the paper ([link](https://openreview.net/forum?id=vL3pmJjGDQ&noteId=H9aievlwc9)), where we highlighted the inherent challenges in evaluating interpretability methods. Specifically:
>
> - Traditional methods often rely on class logits as the primary signal, using strategies such as **supervised semantic segmentation** or **perturbation tests on logits**.
> - Even for explanations of final predictions, it remains unclear whether such approaches faithfully reflect interpretability.
> - Many of these metrics are not directly applicable in our setting due to fundamental methodological differences.
> - Assessing interpretability is inherently challenging; this limitation arises from the nature of interpretability rather than from our methodology or the unsupervised domain.
>
> To address these challenges, we evaluated our framework through **unsupervised semantic segmentation**, **token-level perturbation tests**, and **object selection**, thereby extending the evaluation beyond previous methods.
>
> Regarding **qualitative results**, we have extensively updated figures and captions following discussions with Reviewer T9SA, and added **Figure 4.2** to highlight the contribution of tokens beyond the [CLS] token. These updates are described in **Sections 4.2 and 6 (Limitations)** under **Red Tag 3**.
>
> ---
>
> We would also like to clarify these conceptual points:
>
> - **Goal of ULTra:**
>   ULTra is primarily an *interpretability framework*. **USS** serves both as an application and an evaluation domain.
>
> - **Function $ f(z_i^{(l)}) $:**
>   The choices for \( f \) (sum, energy, weighted projection) are now motivated in more detail (*Section 3.2*, **Blue Tag 2.1.2**).
>
> ---
>
> We again thank Reviewer v9J2 for the thoughtful feedback, which has helped strengthen the clarity, interpretability motivation, and overall presentation of our work.

---

> > ### Author Response · Authors · 2025-10-30
> > **Follow-up on Reviewer Feedback and Clarification Confirmation**
> >
> > Hi, we would greatly appreciate it if you could let us know whether our response has addressed your concerns. Please don’t hesitate to share if there are any remaining points you would like us to clarify or elaborate on further.

---

### Review · Reviewer_iUCf · 2025-10-05

**Summary Of Contributions:**

The authors have provided a thoughtful and largely comprehensive response to the initial review. They have acknowledged the key points and have outlined concrete plans for revision, which is commendable.

However, while the response addresses the "what" of the revisions, several critical aspects concerning the "how" and "why" still require deeper clarification and substantiation in the manuscript itself.

Based on the authors' responses, the following points need to be explicitly and clearly addressed in the revised manuscript to ensure clarity, reproducibility, and robustness of the claims.

1. The authors' response mentions training the projection vectors $w_i$ with "a limited number of batches." For reproducibility, the manuscript must specify: 1) The exact optimizer (e.g., Adam, SGD) and learning rate used; 2) The number of epochs and the batch size; 3) A brief justification or sensitivity analysis for why the chosen number of samples (256) is sufficient and robust across different models (ViT-B, L) and datasets.


2. The authors' response provides a high-level rationale (leveraging all tokens, mapping before aggregation). This discussion must be expanded and empirically grounded in the manuscript: 1) Can you provide a quantitative or qualitative analysis (e.g., in an appendix) that demonstrates the "lower information loss" compared to methods that aggregate first? For instance, visualizing the explanation maps from an aggregated token representation vs. ULTra's approach could be very illuminating; 2) The claim that latent tokens contain richer information than the final CLS token is central to the work. A direct ablation or comparison highlighting this would significantly strengthen the argument.

3. While the authors acknowledge the high cost in the response and Appendix B.2, this is a major practical limitation that deserves a more prominent discussion: 1) The main text (e.g., the conclusion or the segmentation experiment section) should explicitly state this limitation; 2) The promised discussion on "mitigation strategies" (sparse attention, token pruning) should be included. It should be clear whether these were tried in the current work (even if only preliminarily) or are purely speculative for future work. Providing an estimate of the inference time or FLOPs relative to a baseline forward pass would be highly valuable for practitioners.

4. The promised discussion on "mitigation strategies" (sparse attention, token pruning) should be included. It should be clear whether these were tried in the current work (even if only preliminarily) or are purely speculative for future work. Providing an estimate of the inference time or FLOPs relative to a baseline forward pass would be highly valuable for practitioners: 1) Could the authors design a simple human evaluation? For example, asking annotators to identify the most important sentences in a context and measuring the correlation with the top-k tokens highlighted by ULTra? 2) Alternatively, is there a way to leverage existing faithfulness metrics for text explanations, even if adapted, to move beyond pure qualitative demonstration?

5. The authors' defense for using Euclidean distance is reasonable given the class-agnostic nature of the tokens. However, to further bolster this choice: 1) Have the authors experimented with other distance metrics like cosine distance in the embedding space? A brief note on this, even if results were similar, would demonstrate thoroughness; 2) It would be helpful to include a few visual examples in the appendix showing the "positive" and "negative" perturbation regions for a sample of tokens, making the test and its results more concrete for the reader.

6. The authors have committed to grammatical revisions. I strongly advise a thorough proofreading pass, potentially with the help of a native English speaker or professional editing service, to ensure the final manuscript is polished and meets the high presentation standards of TMLR. Please ensure consistent notation (e.g., ULTraWCLIP vs. $ULTra\(_\mathcal{W}^{\text{CLIP}}\)$) throughout the text and figures.

**Audience:**

Yes

**Audience Explanation:**

See Summary Of Contributions.

**Broader Impact Concerns:**

See Summary Of Contributions

**Claims And Evidence:**

Yes

**Claims Explanation:**

See Summary Of Contributions.

**Requested Changes:**

See Summary Of Contributions.

---

> ### Author Response · Authors · 2025-10-08
> **Response to Reviewer iUCf**
>
> ### Authors Response
>
> We sincerely thank the reviewer for their valuable and constructive feedback.
> Following the initial review ([review link](https://openreview.net/forum?id=vL3pmJjGDQ&noteId=j6OU0HbvZL)) and our first response ([response link](https://openreview.net/forum?id=vL3pmJjGDQ&noteId=wA1L0svHcS)), we have thoroughly revised the manuscript.
> All revisions are clearly marked in **blue** throughout the paper, with corresponding **tags** referencing each reviewer comment.
>
> Below is a summary of the major updates:
>
> 1. **Distinction from Prior Work**
>    - In the *Introduction*, **Tag 2.1.1** now highlights the key distinctions between our approach and existing methods.
>
> 2. **Motivation of the Function \( f \)**
>    - In *Section 3.2*, **Tag 2.1.2** includes a detailed explanation and motivation for the design of our function \( f \).
>
> 3. **Table Caption Revisions**
>    - All **table captions** have been updated to explicitly include the *“unsupervised”* tag and expanded explanations for the *“training”* column.
>
> 4. **Explanation of ULTra’s Performance**
>    - The rationale behind **ULTra’s performance** is discussed under **Tag 2.1.4** in *Section 5.2*.
>    - We also added a **new figure (Figure 4.2)** illustrating why relying solely on the [CLS] token can lead to information loss.
>      Specifically, we use an image with multiple objects and animals to demonstrate that tokens beyond [CLS] capture more diverse and informative features. This is explained under **Tag 2.2.2** in *Section 5.2* and in the figure caption.
>
> 5. **Limitations and Complexity Discussion**
>    - The **limitations section** has been moved to the *main text* (*Section 6*) under **Tag 2.1.5**, including potential mitigation strategies and future directions.
>    - Additionally, under **Tag 2.2.3**, we discuss how using **intermediate layers** can reduce time complexity.
>      Supporting experiments are provided in the **Appendix**.
>
> 6. **Clarification on Perturbation Visualizations**
>    - *Figure 5* already includes examples showing both **positive** and **negative** perturbation regions.
>    - We have also tested **cosine similarity**, and the results remain consistent.
>      However, we chose not to include these due to redundancy and the limited contrast caused by cosine similarity’s bounded range (-1–1).
>
> 7. **Additional Text Examples and Metric Update**
>    - Six new **text examples** have been added to the **Appendix**.
>    - An update on the **quantitative evaluation metric** will be provided shortly.
>
> 8. **Improved Grammar and Readability**
>    - The entire manuscript has undergone a **language and style revision** to improve clarity and readability.
>    - We plan to conduct another comprehensive revision once all changes are finalized.
>
> 9. **Societal Impact Discussion**
>    - A dedicated discussion on the **potential negative societal impacts** of our method has been added in *Section 7*.
>
> 10. **Related Work: Post-hoc vs. Ante-hoc Interpretability**
>     - This discussion is included in *Section 2* under **Tag 2.1.10**.
>
> 11. **Training Details and Rationale**
>     - The **training setup and its underlying rationale** have been added to *Section 5.2* under **Tag 2.2.1**.
>
> ---
>
> We once again thank the reviewer for their time and insightful feedback, which have significantly improved the clarity and rigor of our paper.

---

> > ### Author Response · Authors · 2025-10-11
> >
> > Hi, the quantitative evaluation metric has been carefully designed and added, measuring the impact of removing important context tokens on the model’s behavior across different selection rates. The details of the metric, along with experimental results, are presented in **Appendix F.2**.

---

> > > ### Comment · Reviewer_iUCf · 2025-10-25
> > > **Response to Authors**
> > >
> > > After reading the author's response, I still have the following questions:
> > >
> > > (1) The paper explicitly notes the high computational cost of ULTra for segmentation due to the need for multiple backward passes. While this is acknowledged as a limitation, a more quantitative analysis is needed. Providing concrete metrics (e.g., inference time or FLOPs compared to a single forward pass) on a standard dataset would give readers a much clearer understanding of the practical trade-offs.
> > >
> > > (2)  The clustering step is crucial for the segmentation performance. The paper would be strengthened by a more detailed analysis of: The sensitivity of the results to the cluster count k or the threshold $\zeta$; The rationale behind the choice of hierarchical clustering over other methods (e.g., K-Means) and whether the performance is robust to this choice; Appendix E is a good start, but this analysis could be integrated or referenced more prominently in the main paper.
> > >
> > > (3) The three variants of ff are presented and empirically compared. However, the paper lacks a deeper, qualitative analysis of why $f_\mathbf{w}$ learns a better basis or what the learned weights $\mathbf{w}_i$ represent. A brief discussion or visualization of what dimensions are being upweighted could be very insightful.
> > >
> > > (4) The application to LLM interpretability for summarization is a nice demonstration of versatility but remains qualitative. The proposed Comprehensiveness metric (in Appendix F.2) is a step in the right direction, but it should be included and discussed in the main text to provide a quantitative foundation for the claims in the language domain. A comparison with a simple baseline (e.g., attention-based importance) would also strengthen this part.
> > >
> > > (5) In the perturbation strategy for $ULTra_\mathcal{W}$ (Eq. 9), how is the initial segmentation used to define the "negative regions" for perturbation? Does the performance of $ULTra_\mathcal{W}$ depend heavily on the quality of this initial segmentation, and if so, how is instability avoided?
> > >
> > > (6) The method selects a specific layer l for analysis. Did you observe a consistent "sweet spot" layer across different models and tasks? The appendix shows performance across layers, but a summarized guideline or insight on how to choose l in practice would be helpful.
> > >
> > > (7) For the text summarization task, you compute relevance scores for the summary tokens with respect to the context. Have you considered the reverse, computing the influence of the context tokens on the generation of the summary tokens step-by-step, which might be more aligned with the causal nature of text generation?

---

> ### Author Response · Authors · 2025-10-29
> **Response to Reviewer iUCf remaining questions**
>
> We thank the reviewer for the helpful feedback. We provide our responses below.
>
> ---
>
> *1) The paper explicitly notes the high computational cost of ULTra for segmentation due to the need for multiple backward passes. While this is acknowledged as a limitation, a more quantitative analysis is needed. Providing concrete metrics (e.g., inference time or FLOPs compared to a single forward pass) on a standard dataset would give readers a much clearer understanding of the practical trade-offs.*
>
> 1 - Below are the required results.
>
> | Model     | Forward (FLOPs)      | Layer 3         | Layer 5         | Layer 7         | Layer 9         | Layer 11        | Layer 13        |
> |------------|----------------------|-----------------|-----------------|-----------------|-----------------|-----------------|-----------------|
> | ViT-B/32   | 5.7×10¹¹             | 5.6×10¹²        | 2.0×10¹³        | 4.4×10¹³        | 7.7×10¹³        | 1.1×10¹⁴        | 1.7×10¹⁴        |
> | **Ratio**  | 1×                   | 10×             | 35×             | 77×             | 135×            | 193×            | 298×            |
> | ViT-B/16   | 2.3×10¹²             | 9.2×10¹³        | 3.3×10¹⁴        | 7.3×10¹⁴        | 1.2×10¹⁵        | 1.9×10¹⁵        | 2.8×10¹⁵        |
> | **Ratio**  | 1×                   | 40×             | 143×            | 317×            | 521×            | 826×            | 1217×           |
>
> **Table:** FLOPs required to compute a forward pass and to generate gradient-based explanation maps targeting different layers for batch size of 64.
>
> We have also included this information in Appendix B (Tag 2.3.1). ULTRA, originally developed as an interpretability framework, achieves state-of-the-art performance in unsupervised semantic segmentation without requiring any training. As shown in the table above, and as mentioned previously, ULTRA is computationally intensive (particularly in deeper layers), and there remains room for improvement through acceleration of the framework in the future works.
>
> ---
>
> *2) The clustering step is crucial for the segmentation performance. The paper would be strengthened by a more detailed analysis of: The sensitivity of the results to the cluster count k or the threshold ζ; The rationale behind the choice of hierarchical clustering over other methods (e.g., K-Means) and whether the performance is robust to this choice; Appendix E is a good start, but this analysis could be integrated or referenced more prominently in the main paper.*
>
> 2- We have revised Section 4.1 (Tag 2.3.2) to provide a clearer explanation of why hierarchical clustering is particularly well-suited for our method along with a reference to the appendix. We also include a reference to the appendix in the main experiments section for further details.
> Regarding the sensitivity to the threshold ζ, we already present an ablation study in Figure 15 (Appendix G), which demonstrates the robustness of our method for ζ values in the range of 0.2–0.6.
> For the comparison of clustering methods, we had also experimented with K-means. We found that hierarchical clustering achieved slightly better performance (1–2% higher on our evaluation metrics) while offering more intuitive control over segmentation granularity.
>
> ---
>
> *3) The three variants of ff are presented and empirically compared. However, the paper lacks a deeper, qualitative analysis of why ff learns a better basis or what the learned weights represent. A brief discussion or visualization of what dimensions are being upweighted could be very insightful.*
>
> *5) In the perturbation strategy for ULTra-W (Eq. 9), how is the initial segmentation used to define the "negative regions" for perturbation? Does the performance of ULTra-W depend heavily on the quality of this initial segmentation, and if so, how is instability avoided?*
>
> 3,5 - ULTra-W is trained by perturbing regions that are not predicted to belong to the same class as the target token, based on the initial segmentation (see tag 2.3.5 in Section 4.1), and by learning a representation that remains robust under such perturbations. This encourages the model to develop features that are less sensitive to negative regions and more focused on the positive ones. As a result, the generated heatmaps more effectively cover the entire semantics of the target object, compared to those produced by ULTra-S. Qualitative examples and discussion demonstrating these improvements are now provided in Appendix C (tag 2.3.3).
> Although ULTra-W depends on the initial segmentation, we do not expect it to perform worse than the initial result. This is because the target token itself always lies within the positive region, and surrounding areas with positive value are reinforced. In essence, ULTra-W sharpens the heatmap around these positive regions, thereby refining the localization rather than degrading it. Therefore, we do not anticipate any instability issues related to ULTra-W.
>
> ---

---

> > ### Author Response · Authors · 2025-10-29
> >
> > *4) The application to LLM interpretability for summarization is a nice demonstration of versatility but remains qualitative. The proposed Comprehensiveness metric (in Appendix F.2) is a step in the right direction, but it should be included and discussed in the main text to provide a quantitative foundation for the claims in the language domain. A comparison with a simple baseline (e.g., attention-based importance) would also strengthen this part.*
> >
> > *7) For the text summarization task, you compute relevance scores for the summary tokens with respect to the context. Have you considered the reverse, computing the influence of the context tokens on the generation of the summary tokens step-by-step, which might be more aligned with the causal nature of text generation?*
> >
> > 4,7 - The discussion and results are now included in the main text section 5.4 (tag 2.3.4).
> > In our preliminary experiments, we also examined the reverse formulation, namely, computing the influence of context tokens on the generation of summary tokens step by step. However, this approach did not yield informative or distinguishable patterns. The main reason lies in the large imbalance between the number of context tokens and summary tokens in the TL;DR dataset: each summary is relatively short compared to its corresponding context, which is consistent with the nature of the text summarization task. As a result, when aggregating contributions from numerous context tokens (as in Eq. 13), the averaging effect caused these values to become nearly uniform across summary tokens, as local variations tended to cancel out. Consequently, this reverse formulation produced almost constant relevance scores and offered limited interpretive value.
> >
> > ---
> >
> > *6) The method selects a specific layer l for analysis. Did you observe a consistent "sweet spot" layer across different models and tasks? The appendix shows performance across layers, but a summarized guideline or insight on how to choose l in practice would be helpful.*
> >
> > 6 - For a fixed model, the overall performance remains relatively consistent across different datasets, as illustrated in Appendix A Figure 8. This observation suggests the presence of a sweet spot, a region where the model achieves optimal performance. A similar pattern can be observed in Figure 9 of the same appendix, where a flat region in the curve represents this stable performance zone. Interestingly, the location of this sweet spot tends to be similar across datasets for the same model. In practice, it can be identified experimentally by evaluating the model’s performance on a small validation subset and then verifying its consistency on other datasets. This discussion is now added to Appendix A (tag 2.3.6).

---

> > > ### Comment · Reviewer_iUCf · 2025-11-04
> > > **Response to Authors**
> > >
> > > After reading the author's response, I found that the authors had addressed all my concerns.

---

### Comment · Reviewer_T9SA · 2025-08-13
**Strong foundation but unclear application**

The paper proposes **Ultra**, a method designed to provide interpretability to tokens in Visual Transformer architectures. While the method was initially developed for general use, it focuses specifically on unsupervised semantic segmentation (USS). The method is evaluated on multiple datasets (Pascal VOC, COCO-Stuff, Cityscapes, Potsdam) and also reports results in text analysis.

*Side note*: The paper is stated to be of standard length (12 pages), meaning the appendices are not considered mandatory material. If they are intended to be treated as part of the paper, they should be moved into the main text.

**Strengths**:
-    The paper is technically correct: I have checked the mathematics and found no major errors.
-    The results are strong: I have reviewed multiple papers on unsupervised semantic segmentation, and the results presented here are indeed competitive. The paper could stand on its own as a contribution to USS.

**Weaknesses**:
1.    The paper is not particularly clear in its purpose. It oscillates between “interpretability of the token” and unsupervised semantic segmentation. From the abstract and introduction, I initially assumed the focus was on the former, yet the evaluation is centered on USS, with some addition on text. Several aspects remain unclear:
   -    How is the success of “measuring the contribution of latent tokens” evaluated? Since the paper does not explicitly state its main objective, it is unclear: contribution of tokens to what?
   -    The paper states that it measures the influence weight of a token in a specific representation (before Eq. (5)). This is fine, but why is it relevant? Why is it more important than simply aggregating the weights between the two?
2.    What loss function is used for USS training? This is related to the previous point—what is expected from token interpretability? I believe the paper should start with a hand-drawn example illustrating what is expected.
3.    The same issue applies to the text analysis section.
4.    The interpretability results are not compelling. The figures do not communicate much. We see heatmaps (saliency maps) in Figures 1, 3, 4, and 6, but it is unclear what we are supposed to observe and how this demonstrates successful token interpretability. For instance, in Figure 3, the token heatmap focuses on a dog, and then a segmentation mask around the dog is produced. However, the segmentation mask already conveys all the necessary information.


**Concluding weakness**: The paper emphasizes “unsupervised” settings, but interpretability in unsupervised contexts is problematic—since there are no ground truths, it is difficult to judge whether the results are good or not. As a result, the paper lacks a metric for evaluating token contribution or interpretability in representation. To me, the work currently feels like a work in progress rather than a finished piece.

Minor issues:
- text : "experimetns"
- Figure 2 comes in text after figure 3

---

> ### Author Response · Authors · 2025-08-17
>
> Dear Reviewer,
>
> Thank you for your thorough review of the manuscript. We appreciate your recognition of the main contribution of our work (a strong foundation with competitive performance against SOTA methods), and we apologize if the presentation caused confusion. We will incorporate your suggestions to improve clarity.
>
> *1- The paper is not particularly clear in its purpose. It oscillates between “interpretability of the token” and unsupervised semantic segmentation.*
>
> We apologize if our objectives seemed to oscillate. The core research question posed by ULTra is: **Does semantic awareness exist in the latent representations of transformer (TF) models?**
>
> From ULTra heatmaps, we observed clear semantic alignment, prompting a second question:
> **Can this alignment be leveraged for unsupervised semantic segmentation (USS)?**
>
> To address this, we conducted extensive experiments using pretrained TFs for USS, as you noted, achieving results competitive with methods designed explicitly for that task. The **ULTraW** variant further builds on this: it first runs USS with a base ULTra variant, then applies perturbation to learn token-level weights that remain robust under noise—thus deepening our interpretability via USS.
>
> -  *How is the success of “measuring the contribution of latent tokens” evaluated?*
>
> A key challenge in interpretability is the lack of a universal evaluation metric\[1,2\]. In supervised settings, explainability is typically evaluated indirectly via downstream tasks (e.g., semantic segmentation) or perturbation tests—assuming that strong explanation maps (1) improve performance or (2) cause coherent output changes under perturbation.
>
> Since tokens in ULTra naturally have no labels, we extend this principle to the unsupervised domain.
>
> 1. Aggregating token explanation maps and clustering them to form semantic regions for USS.
> 2. Conducting perturbation tests with modifications tailored to our unsupervised context.
> 3. Demonstrating object-level selection: e.g., in Fig. 3, tokens distinctly highlight the dog or cat.
>
> -  *The paper states that it measures the influence weight of a token in a specific representation (before Eq. (5)). This is fine, but why is it relevant? Why is it more important than simply aggregating the weights between the two?*
>
> Tokens are high-dimensional vectors; assuming equal importance across components can obscure critical contributions. We tested three functions: two that aggregate with equal importance, and our proposed ULTraW, which learns the weighting function and generally achieves stronger downstream performance.
>
> *2- What loss function is used for USS training? This is related to the previous point—what is expected from token interpretability?  I believe the paper should start with a hand-drawn example illustrating what is expected.*
>
> Two of our three proposed methods are training-free. The third uses the loss in Eq. (10). The core idea of the loss function is that we want to find the representation projection where it is resilient to noise. This intuitively makes sense because it has been shown that transformers are robust to noise added to the input \[3\]. Regarding expectation from token interpretability and hand-draw example we refer you to our answer for question 4.
>
> *3- Experiments on Text*
>
> The experiments on text data are included to demonstrate that ULTra is not limited to visual TFs but also works in NLP contexts. While not our primary focus, they strengthen the generality claim.
>
> *4- The interpretability results are not compelling. The figures do not communicate much*
>
> We agree that the figures should have been discussed in greater detail. We will update the paper accordingly. For instance, in Fig. 3, tokens clearly separate semantic entities: some attend to the dog, others to the cat, and some even to object attributes (e.g., the cat’s head). In Fig. 4, deeper layers show richer semantic awareness, with tokens gradually capturing the entire elephant. These findings provide evidence that token embeddings encode coherent semantic structure, offering insights into TF internals.
>
> We will also correct the minor issues you pointed out.
>
> Thank you again for your constructive feedback.
>
> Best regards,
> Authors
>
> References:
>
> \[1\] Hila Chefer, Shir Gur, and Lior Wolf. Transformer interpretability beyond attention visualization. Proceed ings of the IEEE/CVF Conference on Computer Vision and Pattern Recognition, pp. 782–791, 2021b.
>
> \[2\] Junyi Wu, Bin Duan, Weitai Kang, Hao Tang, and Yan Yan. Token Transformation Matters: Towards Faithful Post-Hoc Explanation for Vision Transformer . In 2024 IEEE/CVF Conference on Computer Vision and Pattern Recognition (CVPR), Los Alamitos, CA, USA, June 2024\. IEEE Computer Society.
>
> \[3\] Zhou, D., Xu, C., Zhang, S., & Tao, D. (2022). *Understanding the Robustness in Vision Transformers*. In International Conference on Machine Learning (ICML), pp. 27378–27394. PMLR.

---

### Author Response · Authors · 2025-08-22
**Requested Changes from reviewer T9SA**

Following up on our earlier response (see [OpenReview comment](https://openreview.net/forum?id=vL3pmJjGDQ&noteId=q1tUkJpMUD)),
we have updated the paper with the following revisions to address the raised concerns.
We thank reviewer T9SA for the constructive comments, which helped us improve the clarity and completeness of the paper.

**Q1. Clarification of objective and role of USS**
- In the *Introduction*, we now explicitly state the main objective of the paper and clarify the role of USS.
- In *Section 4.2*, we expanded the discussion of the evaluation and elaborated on why our approach is more meaningful than simply aggregating weights (before Eq. 5).

**Q2. Loss function of UltraW**
- In *Section 3.2(iii)* and *Section 4.1*, we added a more detailed discussion of the UltraW loss function, including clarifications before Eq. 9.

**Q3. Text section**
- In *Section 5.4 (last paragraph)*, we highlight that the text-data experiments demonstrate ULTra’s applicability beyond visual TFs, extending also to NLP contexts.  While NLP is not the primary focus of the paper, these results strengthen the generality claim.

**Q4. Figures and discussion**
- We enhanced the discussions of Figures 1, 2, and 4 in both their captions and the main text.
- Additional commentary is included in the *Introduction* (paragraph 4) and in *Section 4.2* (after the reference to Figure 4).

---

### Comment · Reviewer_iUCf · 2025-09-07
**Reviews**

The paper presents a novel framework, ULTra, for interpreting latent tokens in Transformers and demonstrates its utility in unsupervised semantic segmentation and model interpretability. The work is timely, well-motivated, and empirically validated across multiple datasets and tasks. However, several aspects require clarification, expansion, or correction to meet the publication standards of TMLR.

(1) The contributions are listed but could be more clearly differentiated from prior work. Please explicitly state how ULTra advances beyond methods like Chen et al. (2024) or attention-based explainability techniques.

(2) Equation (4) introduces f(z _i^(l)) without explicit definition. Please clarify the choice of f and its role in the gradient computation. The three variants (f _s, f_e, f_w) are introduced later but should be motivated earlier.

(3) Tables 1–4 include many baselines, but it is unclear which methods are unsupervised and which require supervision or fine-tuning. Please add a brief note in the caption or text to clarify this.

(4) ULTra outperforms these methods, the paper should discuss why a training-free method outperforms those that require fine-tuning. Is it due to better token semantics or the aggregation method?

(5) The paper mentions high computational cost in Appendix B.2. This is a significant limitation and should be discussed in the main text, along with potential mitigation strategies.

(6) The perturbation test and object selection are good ideas, but the metrics (e.g., Euclidean distance in Table 5) are not standardized. Consider using established metrics like Insertion/Deletion AUC or similar.

(7) The text summarization experiment is qualitative and limited to two examples. To strengthen the claim of cross-modal applicability, include more examples or a quantitative evaluation.

(8) There are minor grammatical errors and long sentences that affect readability. Consider revising for clarity.

(9) The paper should briefly discuss potential negative societal impacts or limitations of the method, especially regarding interpretability of LLMs and possible misuse.

(10) The related work section is comprehensive but could be better organized. Consider grouping methods by category (e.g., post-hoc vs. ante-hoc, segmentation-specific vs. general interpretability).

---

> ### Author Response · Authors · 2025-09-19
> **Respond to reviewer  iUCf**
>
> We sincerely thank reviewer iUCf for the constructive and detailed feedback, as well as for recognizing the novelty of our work. Below, we provide point-by-point responses and outline the planned revisions.
>
> **(1) Differentiation from prior work**
> Prior explainability methods for Transformers typically emphasize attributing predictions to input tokens or visual regions, often focusing solely on the final class decision (e.g. Chefer et al (2021) and Wu et all(2024)). These approaches do not directly reveal the semantics of intermediate token representations. The most related work, Chen et al. (2024), generates explanations by aligning with a text modality and disabling cross-attention in subsequent layers. However, this requires altering the model architecture, which introduces distribution shift and may reduce reliability. In contrast, ULTra (i) provides a direct interpretation of latent tokens without relying on any aligned modality, and (ii) preserves the model entirely at inference time, avoiding modifications that could compromise faithfulness. While these distinctions are mentioned in different parts of the paper, we will make them more explicit in the introduction and contributions section to clearly delineate our advances beyond prior work.
>
> **(2) Definition and motivation of f**
>  We will revise Equation (4) to explicitly define f at the point, clarify its role in gradient computation, and provide earlier motivation for the three variants ($f_s,f_e,f_w$).
>
> **(3) Clarification of supervision in baselines**
>  All benchmarks in Tables 1–4 are unsupervised, consistent with ULTra. The “training” column refers to whether baselines rely on pre-trained models with or without fine-tuning. We will explicitly note this in the table captions to avoid confusion.
>
> **(4) Why ULTra outperforms fine-tuned methods**
>  We identify two main reasons why ULTra performs strongly despite being training-free:
>  (i) ULTra leverages information across *all* latent tokens, which is often richer than relying only on the final token representation.
>  (ii) ULTra maps multiple embeddings into the input space *before* aggregation, leading to lower information loss compared to prior methods that aggregate first and then map.
>  We will add this discussion to the main text.
>
> **(5) Computational cost**
>  We acknowledge the high computational complexity of our method. While Appendix B.2 provides details, we will integrate this limitation into the main text and discuss mitigation strategies such as approximate sparse attention methods, token pruning, and parallelization. We will also note that while our approach is applicable to unsupervised semantic segmentation, its primary goal is to provide insight into Transformer representations.
>
> **(6) Perturbation metrics**
> Our perturbation analysis in Table 5 serves as a sanity check on latent token behavior, quantifying embedding shifts under perturbations. While established metrics such as Insertion/Deletion AUC are well-suited for class-specific saliency maps, they do not directly apply to our class-agnostic token maps. In Table 5, the comparisons are conducted within the token space, where the straightforward application of such metrics is not feasible. We therefore adopted the Euclidean norm, which captures the same underlying idea of measuring perturbation sensitivity. Importantly, the results show that even with this naive choice, positive perturbations consistently induce more substantial alterations in token representations than negative ones.
>
> **(7) Text summarization experiments**
> We acknowledge that our current evaluation on text data is limited, primarily due to the lack of standard datasets. In response, we will strengthen our analysis by including additional qualitative examples and expanding the discussion
>
> **(8) Grammar and readability**
>  We will revise the manuscript carefully to fix grammatical issues, shorten overly long sentences, and improve overall readability.
>
> **(9) Societal impact and limitations**
>  We will add a discussion on potential negative societal impacts, particularly the risks of misinterpretation or over-reliance on LLM explanations, as well as broader limitations such as computational cost and metric choice.
>
> **(10) Organization of related work**
> We already group related work into interpretability, unsupervised semantic segmentation, and latent embedding–based methods. We will expand the interpretability section to discuss ante-hoc approaches as well, making the distinction from our post-hoc method clearer.
>
> Thank you again for your constructive and insightful comments.
>
> Best regards,
>
> Authors

---

> > ### Comment · Reviewer_iUCf · 2025-10-01
> > **Repaly to authors**
> >
> > The authors have provided a thoughtful and largely comprehensive response to the initial review. They have acknowledged the key points and have outlined concrete plans for revision, which is commendable.
> >
> > However, while the response addresses the "what" of the revisions, several critical aspects concerning the "how" and "why" still require deeper clarification and substantiation in the manuscript itself.
> >
> > Based on the authors' responses, the following points need to be explicitly and clearly addressed in the revised manuscript to ensure clarity, reproducibility, and robustness of the claims.
> >
> > 1. The authors' response mentions training the projection vectors $w_i$ with "a limited number of batches." For reproducibility, the manuscript must specify: 1) The exact optimizer (e.g., Adam, SGD) and learning rate used; 2) The number of epochs and the batch size; 3) A brief justification or sensitivity analysis for why the chosen number of samples (256) is sufficient and robust across different models (ViT-B, L) and datasets.
> >
> >
> > 2. The authors' response provides a high-level rationale (leveraging all tokens, mapping before aggregation). This discussion must be expanded and empirically grounded in the manuscript: 1) Can you provide a quantitative or qualitative analysis (e.g., in an appendix) that demonstrates the "lower information loss" compared to methods that aggregate first? For instance, visualizing the explanation maps from an aggregated token representation vs. ULTra's approach could be very illuminating; 2) The claim that latent tokens contain richer information than the final CLS token is central to the work. A direct ablation or comparison highlighting this would significantly strengthen the argument.
> >
> > 3. While the authors acknowledge the high cost in the response and Appendix B.2, this is a major practical limitation that deserves a more prominent discussion: 1) The main text (e.g., the conclusion or the segmentation experiment section) should explicitly state this limitation; 2) The promised discussion on "mitigation strategies" (sparse attention, token pruning) should be included. It should be clear whether these were tried in the current work (even if only preliminarily) or are purely speculative for future work. Providing an estimate of the inference time or FLOPs relative to a baseline forward pass would be highly valuable for practitioners.
> >
> > 4. The promised discussion on "mitigation strategies" (sparse attention, token pruning) should be included. It should be clear whether these were tried in the current work (even if only preliminarily) or are purely speculative for future work. Providing an estimate of the inference time or FLOPs relative to a baseline forward pass would be highly valuable for practitioners: 1) Could the authors design a simple human evaluation? For example, asking annotators to identify the most important sentences in a context and measuring the correlation with the top-k tokens highlighted by ULTra? 2) Alternatively, is there a way to leverage existing faithfulness metrics for text explanations, even if adapted, to move beyond pure qualitative demonstration?
> >
> > 5. The authors' defense for using Euclidean distance is reasonable given the class-agnostic nature of the tokens. However, to further bolster this choice: 1) Have the authors experimented with other distance metrics like cosine distance in the embedding space? A brief note on this, even if results were similar, would demonstrate thoroughness; 2) It would be helpful to include a few visual examples in the appendix showing the "positive" and "negative" perturbation regions for a sample of tokens, making the test and its results more concrete for the reader.
> >
> > 6. The authors have committed to grammatical revisions. I strongly advise a thorough proofreading pass, potentially with the help of a native English speaker or professional editing service, to ensure the final manuscript is polished and meets the high presentation standards of TMLR. Please ensure consistent notation (e.g., ULTraWCLIP vs. $ULTra\(_\mathcal{W}^{\text{CLIP}}\)$) throughout the text and figures.

---

### Comment · Reviewer_iUCf · 2025-10-04
**Repaly to authors**

The authors have provided a thoughtful and largely comprehensive response to the initial review. They have acknowledged the key points and have outlined concrete plans for revision, which is commendable.

However, while the response addresses the "what" of the revisions, several critical aspects concerning the "how" and "why" still require deeper clarification and substantiation in the manuscript itself.

Based on the authors' responses, the following points need to be explicitly and clearly addressed in the revised manuscript to ensure clarity, reproducibility, and robustness of the claims.

1. The authors' response mentions training the projection vectors $w_i$ with "a limited number of batches." For reproducibility, the manuscript must specify: 1) The exact optimizer (e.g., Adam, SGD) and learning rate used; 2) The number of epochs and the batch size; 3) A brief justification or sensitivity analysis for why the chosen number of samples (256) is sufficient and robust across different models (ViT-B, L) and datasets.


2. The authors' response provides a high-level rationale (leveraging all tokens, mapping before aggregation). This discussion must be expanded and empirically grounded in the manuscript: 1) Can you provide a quantitative or qualitative analysis (e.g., in an appendix) that demonstrates the "lower information loss" compared to methods that aggregate first? For instance, visualizing the explanation maps from an aggregated token representation vs. ULTra's approach could be very illuminating; 2) The claim that latent tokens contain richer information than the final CLS token is central to the work. A direct ablation or comparison highlighting this would significantly strengthen the argument.

3. While the authors acknowledge the high cost in the response and Appendix B.2, this is a major practical limitation that deserves a more prominent discussion: 1) The main text (e.g., the conclusion or the segmentation experiment section) should explicitly state this limitation; 2) The promised discussion on "mitigation strategies" (sparse attention, token pruning) should be included. It should be clear whether these were tried in the current work (even if only preliminarily) or are purely speculative for future work. Providing an estimate of the inference time or FLOPs relative to a baseline forward pass would be highly valuable for practitioners.

4. The promised discussion on "mitigation strategies" (sparse attention, token pruning) should be included. It should be clear whether these were tried in the current work (even if only preliminarily) or are purely speculative for future work. Providing an estimate of the inference time or FLOPs relative to a baseline forward pass would be highly valuable for practitioners: 1) Could the authors design a simple human evaluation? For example, asking annotators to identify the most important sentences in a context and measuring the correlation with the top-k tokens highlighted by ULTra? 2) Alternatively, is there a way to leverage existing faithfulness metrics for text explanations, even if adapted, to move beyond pure qualitative demonstration?

5. The authors' defense for using Euclidean distance is reasonable given the class-agnostic nature of the tokens. However, to further bolster this choice: 1) Have the authors experimented with other distance metrics like cosine distance in the embedding space? A brief note on this, even if results were similar, would demonstrate thoroughness; 2) It would be helpful to include a few visual examples in the appendix showing the "positive" and "negative" perturbation regions for a sample of tokens, making the test and its results more concrete for the reader.

6. The authors have committed to grammatical revisions. I strongly advise a thorough proofreading pass, potentially with the help of a native English speaker or professional editing service, to ensure the final manuscript is polished and meets the high presentation standards of TMLR. Please ensure consistent notation (e.g., ULTraWCLIP vs. $ULTra\(_\mathcal{W}^{\text{CLIP}}\)$) throughout the text and figures.

---

### Author Response · Authors · 2025-11-03
**Summary of Rebuttal**

We sincerely thank all reviewers for their thoughtful engagement and constructive feedback on our submission, *“ULTra: Unveiling Latent Token Interpretability in Transformer-Based Understanding and Segmentation.”* We are grateful for their valuable insights, which have greatly helped us improve the clarity, completeness, and overall quality of the paper.

Our paper introduces **ULTra**, a framework for interpreting latent token representations in Transformer-based models and uncovering their semantic structure. ULTra enables unsupervised semantic segmentation directly from pre-trained models without fine-tuning and further improves segmentation via a self-supervised refinement that learns an external transformation matrix. It achieves state-of-the-art results in unsupervised segmentation and demonstrates broad applicability across tasks such as object selection and interpretable text summarization, highlighting its value for understanding Transformer embeddings.

It is important to emphasize that we have not altered our core methodology or claims during the rebuttal period. Instead, our efforts focused on clarifying our original contributions, improving the presentation and interpretability of results, and extending empirical analyses to address the reviewers’ constructive suggestions.

Here are the key improvements and clarifications made based on reviewers’ feedback:

---

### **1. Clarification of the Paper’s Goals and Evaluation Process**
We substantially revised the *Introduction* section to better articulate ULTra’s main objectives, its connection to the (USS) pipeline, and how it differs from previous interpretability studies. We also refined the evaluation methodology and expanded the discussion of interpretability challenges (Section 4.2 and the Conclusion), emphasizing that such challenges are shared even among simpler prior works that primarily analyze final class logits.
*(Addressing Reviewers T9SA, v9J2, and iUCf)*

---

### **2. Additional Qualitative Results and Figure Analysis**
All figure captions have been revised for better contextual explanation, and we expanded the corresponding text discussions to clarify the interpretation and significance of each visualization. Furthermore, we added **Figure 4 (Example 2)** presenting a complex, multi-object example to highlight how tokens beyond CLS can capture fine-grained contextual information. Additional qualitative results and analyses have also been included in **Appendix C** for deeper insight of ULTraw.
*(Addressing Reviewers T9SA)*

---

### **3. Limitations and Computational Complexity**
We moved the Limitations section from the appendix into the main paper to make it more visible and transparent. Moreover, we added **Appendix B** that quantifies ULTra’s computational complexity for segmentation tasks and discusses potential directions for future optimization.
*(Addressing Reviewer iUCf and v9J2)*

---

### **4. Expanded Text Experiments and Quantitative Evaluation**
The text experiment section was expanded with more examples, additional discussion, and a newly introduced *quantitative metric*, complementing the qualitative results, which are presented in **Appendix H**. This enhances the robustness of our evaluation and demonstrates ULTra’s interpretability in language-related tasks as well.
*(Addressing Reviewers iUCf and T9SA)*

---

### **5. Other Revisions and General Improvements**
We incorporated several additional refinements throughout the paper:
- Strengthened motivation for different ULTra variants and clarified the rationale for the hierarchical segmentation design.
- Added a dedicated **Social Impact** section.
- Expanded training details for transparency and reproducibility.
- Carefully revised grammar and style across the paper for clarity and readability.

---

We are confident that these revisions and clarifications have substantially strengthened the paper in both presentation and technical depth, while reinforcing the validity and importance of our original contributions.

Thank you again for your time, effort, and constructive feedback.

Sincerely,
The Authors

---

### Author Response · Authors · 2025-11-14
**Update?**

Hi,

It has been several weeks since the discussion period started (5 Oct). We wanted to check if there are any updates on the decision or if there is anything you might need from us.

---

### Decision · Action_Editor_74UV · 2025-12-11

**Recommendation:** Accept with minor revision

**Additional Comments:**

[1] A high-level summary of the proposed method should be included in the introduction, as this is a technical paper and would benefit from a concise overview of the methodology.

[2] Will you release the code to ensure reproducibility? If yes, the paper should include a link to the repository.

**Audience:**

Yes

**Audience Explanation:**

Researchers who work on computer vision may find this paper helpful.

**Claims And Evidence:**

Yes

**Claims Explanation:**

The paper introduces ULTra, a framework designed to describe latent tokens in Transformer models by analyzing the semantic contribution of individual tokens. The framework can be applied to pre-trained models without requiring fine-tuning and has been evaluated on multiple vision datasets, along with an additional qualitative test on text summarization. While the approach may not be entirely novel, this is a well-executed and thoughtfully presented paper.